# STEALIX: MODEL STEALING VIA PROMPT EVOLUTION

## ABSTRACT

Model stealing poses a significant security risk in machine learning by enabling attackers to replicate a black-box model without access to its training data, thus jeopardizing intellectual property and exposing sensitive information. Recent methods that use pre-trained diffusion models for data synthesis improve efficiency and performance but rely heavily on manually crafted prompts, limiting automation and scalability, especially for attackers with little expertise. To assess the risks posed by open-source pre-trained models, we propose a more realistic threat model that eliminates the need for prompt design skills or knowledge of class names. In this context, we introduce Stealix, the first approach to perform model stealing without predefined prompts. Stealix uses two open-source pre-trained models to infer the victim model's data distribution, and iteratively refines prompts through a genetic algorithm based on a proxy metric, progressively improving the precision and diversity of synthetic images. Our experimental results demonstrate that Stealix significantly outperforms other methods, even those with access to class names or fine-grained prompts, while operating under the same query budget. These findings highlight the scalability of our approach and suggest that the risks posed by pre-trained generative models in model stealing may be greater than previously recognized.

## 1 INTRODUCTION

Model stealing allows attackers to replicate the functionality of machine learning models without direct access to training data or model weights. By querying the victim model with hold-out datasets, the attacker can construct a proxy model that behaves similarly to the original by mimicking its predictions. This type of attack compromises the model owner's intellectual property and may expose sensitive information contained in the model, posing both security and privacy risks (Beetham et al., 2022; Carlini et al., 2024).

Current model stealing methods for image classification can be categorized based on the source of the queried images: (1) using publicly available images (Orekondy et al., 2019; Zhao et al., 2024), (2) generating images by training a generator within Generative Adversarial Networks (GANs) from scratch (Truong et al., 2021; Sanyal et al., 2022), or (3) synthesizing images by prompting pre-trained open-source generative models (Shao et al., 2023; Hondru & Ionescu, 2023). The latter uses models like Stable Diffusion (Rombach et al., 2022) to achieve superior efficiency by reducing the dependence on online data sources and by eliminating the high computational cost of training new generators. Previous approaches use human-crafted prompts or class names to synthesize images with a text-to-image model, but they overlook scenarios where class names lack context or fail to capture victim data features. Attackers may also lack sufficient knowledge of the victim's data distribution or may struggle to describe it accurately. Moreover, the dependence on human intervention can greatly hinder scalability and automation, thus limiting the applicability of model stealing. These issues are most prevalent in specialized fields, where the highest value models can be found (e.g., medical applications). Therefore, existing research under the current assumptions may oversimplify the problem and underestimate the threat posed by model stealing enhanced by pre-trained models.

To address these limitations and assess the risk of pre-trained models in model stealing, we propose a realistic threat model where the attacker lacks prior knowledge or expertise in designing prompts

Figure 1: Given a real image as seed and the API access to a black-box victim model, Stealix infers the implicit concept and synthesizes images by iteratively optimizing the prompt with the victim model's response. The attacker can then use these synthetic images to train a proxy model.

for the victim's data. Despite being more realistic, this setup presents a significant challenge to existing methods that rely on pre-trained open-source models, as they depend heavily on prompt design. Without prior knowledge, attackers struggle to create effective prompts that capture the class information learned by the victim model and ensure diversity in query data, thus limiting their ability to steal the model efficiently.

In this work, we introduce Stealix, the first model stealing approach that removes the need for human-crafted prompts by leveraging two pre-trained open-source models, as depicted in Figure 1. Our method employs a text-to-image generative model and a vision-language model to iteratively generate multiple refined prompts for each class. We employ contrastive learning to optimize the prompt to describe the target class based on features extracted from the prompt itself and from image triplets by the vision-language model. To further improve the precision and diversity of the prompts, we propose a proxy metric as the fitness function to evaluate and evolve the prompts. In practice, our approach requires only a single real image per class. We show that this is sufficient to achieve superior performance without requiring manual prompt engineering. We emphasize that this assumption is practical, as potential attackers, typically competitors, often have limited data available, but fail to synthesize more. Overall, we summarize our contributions as follows.

**Contributions.** (i) We introduce a more practical threat model that eliminates the need for expertise in prompt design and closely reflects scalability needs in real-world scenarios. (ii) We propose Stealix, the first prompt-unknown approach that leverages a proxy metric to iteratively refine the prompts. The statistical analysis demonstrates a high correlation between the proxy metric and the feature distance to the victim data. (iii) Our method outperforms approaches that rely on class names or human-designed prompts across multiple datasets, an assumption frequently not held in practice. It achieves up to a 22.2% improvement in attacker model accuracy at a low query budget. (iv) We expose significant risks associated with open-source models in model stealing, highlighting the urgent need for advanced defenses or strategies to prevent their harmful exploitation.

## 2 RELATED WORKS

**Knowledge distillation.** Knowledge distillation (KD) is a model compression technique that trains a smaller student model to replicate the performance of a larger teacher model, thereby enabling deployment on hardware with limited computational resources (Ba & Caruana, 2014; Hinton et al., 2015). Traditional KD methods assume access to the teacher's training data, allowing the student to learn from the same data distribution. When this is impractical due to data size or sensitivity, alternatives such as surrogate datasets (Lopes et al., 2017) or data-free KD using data generators (Fang et al., 2019; Micaelli & Storkey, 2019) are employed. These methods typically require white-box access to the teacher model for back-propagation. In contrast, model stealing adopts an adversarial approach, where neither training data nor internal model details are available to the attacker.

**Model Stealing.** Model stealing aims to replicate either the victim model's properties, such as hyperparameters or learned parameters (Wang & Gong, 2018; Tramèr et al., 2016), or its behavior, known as functionality model stealing (Oliynyk et al., 2023). The latter involves training a proxy model to mimic the victim's behavior and is a threat across domains, including images (Truong et al., 2021), language (Krishna et al., 2020), and robotics (Zhuang et al., 2024). Our work falls into the second category in the image domain. Prevalent methods train a generator from scratch

to adversarially synthesize data for querying the victim model (Truong et al., 2021; Sanyal et al., 2022; Beetham et al., 2022), but this demands millions of queries, making it costly. Recently, more efficient methods using pre-trained diffusion models have been proposed, reducing query budget and enhancing model stealing performance (Shao et al., 2023; Hondru & Ionescu, 2023). For instance, Active Self-Paced Knowledge Distillation (ASPKD) (Hondru & Ionescu, 2023) uses a three-step process: generating images with a diffusion model, querying the victim model with a limited set, and pseudo-labeling the remaining samples using nearest neighbors for attacker model training. However, these methods depend on class-name prompts, which are inadequate for generating images that are similar to the victim data in complex scenarios. To address this, we propose a method that begins with a randomly initialized prompt and refines it iteratively, effectively removing the reliance on such prior knowledge.

**Textual inversion.** Textual inversion (Gal et al., 2023) is a method that learns a prompt corresponding to a specific image or set of images, enabling pre-trained text-to-image models to generate personalized and more targeted outputs. One notable application is DA-Fusion (Trabucco et al., 2024), which employs textual inversion to learn prompts from a seed image and synthesize similar images for data augmentation. This approach shows potential for our threat model, where the attacker has a seed image but lacks the prompt needed to generate relevant images. To assess the effectiveness of DA-Fusion in our threat model, we adopt it as a baseline by replacing the original labels with the victim model's predictions for attacker model training.

## 3 THREAT MODEL

In this section, we formalize the proposed practical threat model for model stealing. We begin by introducing the necessary notations and definitions. Next, we describe the behavior of the victim model. Finally, we detail the attacker's goals and available knowledge, highlighting the constraints that make model stealing challenging.

**Notations.** Let $\mathcal{D} = \{(\boldsymbol{x}_i, y_i)\}$ be the dataset used to train an image classification model, where $\boldsymbol{x}_i \in \mathbb{R}^{H \times W \times C}$ represents input images with height $H$, width $W$, and $C$ channels, and $y_i \in \{1, 2, \ldots, K\}$ denotes the corresponding class labels, with $K$ being the total number of classes. Each class is indexed by $c \in \{1, 2, \ldots, K\}$. The pre-trained generative model $G$ generates an image $\boldsymbol{x} \sim G(\boldsymbol{p}, \epsilon)$ from a given prompt $\boldsymbol{p}$ by denoising noise $\epsilon$ drawn from a standard normal distribution $\epsilon \sim \mathcal{N}(0, 1)$. For brevity, we denote this process as $\boldsymbol{x} \sim G(\boldsymbol{p})$.

**Victim model.** The victim trains a classification model $V$ with parameters $\theta_v$ on a dataset $\mathcal{D}_V$, where images are drawn from the victim data distribution $\boldsymbol{x} \sim \mathcal{P}_V$. Once deployed, it operates as a black-box accessible for queries. To mitigate model stealing risks, we consider the victim model returns only the top-1 predicted class, effectively reducing the amount of information available to potential attackers (Sanyal et al., 2022). Thus, for a given input image $\boldsymbol{x}$, $y^* = V(\boldsymbol{x}; \theta_v) \in \{1, 2, \ldots, K\}$ is the predicted class label.

**Goal and knowledge of the attacker.** The attacker's objective is to train a surrogate model $A(\boldsymbol{x}; \theta_a)$, parameterized by $\theta_a$ that replicates the behavior of the victim model $V$. The attacker has black-box access to $V$, allowing them to query the model with images and receive the predicted top-1 labels. The attacker is constrained by a query budget, representing the total number of queries available per class, denoted as $B$. The attacker lacks knowledge of (i) the architecture and parameters of $V$, (ii) the dataset $\mathcal{D}_V$ used to train $V$, and (iii) prompt design expertise. We also limit the use of class names, as they may by chance serve as good prompts; using them would diverge from the assumption that the attacker lacks prompt design expertise. The lack of the prompt design expertise significantly limits the attacker to leverage the generative model for efficient model stealing.

## 4 APPROACH: STEALIX

This section introduces the details of Stealix. We formalize the problem and give the overview of the method in Section 4.1, followed by an explanation of each of its components (Sections 4.2 to 4.4).

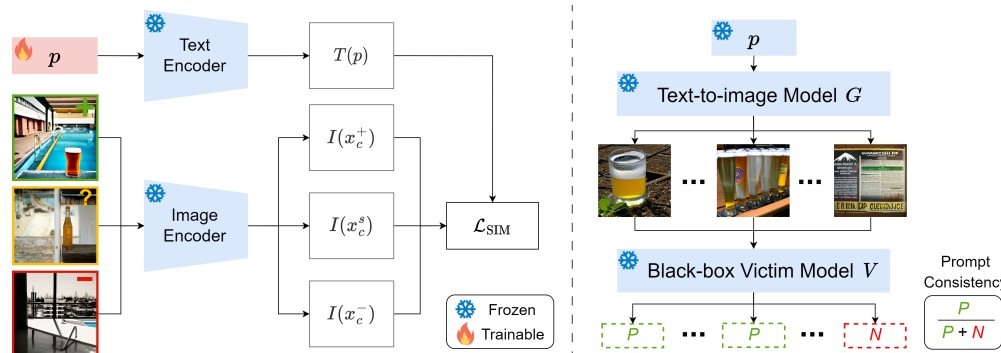

Figure 2: Prompt refinement (left) optimizes the prompt $p$ using encoders $T$ and $I$ via Equation (3) to capture features from seed image $x^s$ and positive image $x^+$ while filtering out negatives from $x^-$. Prompt consistency (right) evaluates $p$ with Equation (5) by prompting generative model $G$ to synthesize images, which are classified by the victim model $V$ to update positive and negative sets. In this example, the negative feature "pool" is removed from the prompt for class "bottle".

### 4.1 METHOD OVERVIEW

The attacker's goal is to optimize the parameters $\theta_a$ of a surrogate model $A$ to replicate the behavior of the victim model $V$ on the victim data distribution $\mathcal{P}_V$, achieving comparable performance. This can be expressed by minimizing the loss between the outputs of the victim and surrogate models over the victim's data distribution, for this task we consider cross-entropy loss:

$$\arg\min_{\theta_a} \mathbb{E}_{x \sim \mathcal{P}_V} \left[ \mathcal{L}_{\text{CE}} \left( V(x), A(x) \right) \right] \tag{1}$$

Without access to the victim data distribution, previous methods (Shao et al., 2023; Hondru & Ionescu, 2023) turn to generate high-quality images using a pre-trained text-to-image model $G$ with a prompt $\mathbf{p}$. By designing prompts to synthesize images similar to the victim data, the attacker can effectively steal the model by minimizing loss on these generated images:

$$\arg\min_{\theta_a} \mathbb{E}_{x \sim G(p)} \left[ \mathcal{L}_{\text{CE}} \left( V(x), A(x) \right) \right] \tag{2}$$

In practice, attackers lack the expertise to design effective prompts, making model stealing challenging. To address this, we propose **Stealix**, a model stealing method built upon genetic algorithm (Zames, 1981). Stealix iteratively generates multiple prompts that capture diverse class-specific features recognized by the victim model, thereby enhancing the efficiency of model stealing. For each class $c$, we define three image sets: the seed set $X_c^s = \{ x_c^s \mid V(x_c^s) = c \}$ of real images classified as $c$ by the victim model; the initially empty positive set $\mathcal{X}_c^+ = \{ x_c^+ \mid V(x_c^+) = c \}$ for synthetic images classified as $c$; and the initially empty negative set $\mathcal{X}_c^- = \{ x_c^- \mid V(x_c^-) \neq c \}$ for those classified as others. We initialize the population $\mathcal{S}^t = \{ (x_c^s, x_c^+, x_c^-)_i^t \}_{i=1}^N$, at generation $t$, consisting of $N$ image triplets, where each component is randomly sampled from $X_c^s$, $\mathcal{X}_c^+$, and $\mathcal{X}_c^-$, respectively. Each triplet is used to optimize a randomly initialized prompt through **prompt refinement** to capture target class features. We evaluate the refined prompts using **prompt consistency**, a fitness metric based on how consistently the victim model classifies synthesized images as the target class, then add the synthetic images to $\mathcal{X}_c^+$ and $\mathcal{X}_c^-$. After evaluating fitness, we perform **prompt reproduction** to generate the next population $\mathcal{S}^{t+1}$, producing diverse triplets and the associated prompts that better capture class-specific features.

We iteratively perform **prompt refinement**, triplet evaluation via **prompt consistency**, and **prompt reproduction** until the query budget $B$ per class is exhausted ($|\mathcal{X}_c^+| + |\mathcal{X}_c^-| = B$). Across $K$ classes, this produces $K \times B$ synthetic images, which, along with seed images, are used to train the attacker model. We operate with $|\mathcal{X}_c^s| = 1$, reducing the attacker's initial input requirements. The complete method is outlined in Algorithm 1 and detailed below.

## 4.2 PROMPT REFINEMENT

Efficient model stealing requires synthesizing images that are similar to the victim data, necessitating prompts that capture the class-specific features learned by the victim model. To achieve this, we optimize the prompt to emphasize attributes leading to correct classifications while minimizing misleading features that cause incorrect predictions, with a triplet of images $\mathbb{X}_c^{s+-} = \{\boldsymbol{x}_c^s, \boldsymbol{x}_c^+, \boldsymbol{x}_c^-\}$. This triplet, along with a random prompt, is projected into a shared feature space using an image encoder $I$ and a text encoder $T$ from a pre-trained vision-language model (Figure 2 left). The prompt is optimized by minimizing the average similarity loss between the prompt and image features, with guidance from the victim model's predictions:

$$\min_p \frac{1}{|\mathbb{X}_c^{s+-}|} \sum_{\mathbf{x} \in \mathbb{X}_c^{s+-}} \mathcal{L}_{\text{SIM}}(I(\boldsymbol{x}), T(\boldsymbol{p}), V(\boldsymbol{x})) \tag{3}$$

where the similarity loss $\mathcal{L}_{\text{SIM}}$ is defined as:

$$\mathcal{L}_{\text{SIM}}(I(\boldsymbol{x}), T(\boldsymbol{p}), V(\boldsymbol{x})) = \begin{cases} 1 - \cos(I(\boldsymbol{x}), T(\boldsymbol{p})), & \text{if } V(\boldsymbol{x}) = c \\ \max(0, \cos(I(\boldsymbol{x}), T(\boldsymbol{p}))), & \text{if } V(\boldsymbol{x}) \neq c \end{cases} \tag{4}$$

We adopt the hard prompt optimization method proposed by Wen et al. (2024) (see Algorithm 2 in Appendix A). This refinement process ensures that the prompt highlights attributes essential for accurate classification while eliminating features that may lead to misclassifications.

## 4.3 PROMPT CONSISTENCY

To evaluate whether the optimized prompt effectively captures the features learned by the victim model, we propose a proxy metric, prompt consistency (PC). Since direct access to the victim data distribution is unavailable, this metric serves as an indicator of distribution similarity and is used for prompt reproduction. We assume that if a prompt captures the latent features of the target class learned by the victim model, the synthetic images will be consistently classified as the target class by the victim model, implying a closer resemblance with the victim data. Based on this assumption, PC measures how well a prompt generates images that match the target class $c$ (Figure 2 right). Given a prompt $\boldsymbol{p}$, a batch of synthetic images $\{\boldsymbol{x}_i\}_{i=1}^M \sim G(\boldsymbol{p})$ is generated, where $M$ is the number of images. PC is computed as:

$$\text{PC} = \frac{1}{M} \sum_{i=1}^M \mathbb{I}(V(\boldsymbol{x}_i) = c) \tag{5}$$

where $\mathbb{I}(V(\boldsymbol{x}_i) = c)$ is 1 if the victim model classifies $\boldsymbol{x}_i$ as class $c$, and 0 otherwise. In Section 5.2, we show there is a strong correlation between PC and the $L_2$ distance between the mean feature vectors of real and generated images, validating PC as an effective proxy metric for monitoring data similarity and for prompt reproduction. The synthetic images are also used to update the image sets $\mathcal{X}_c^+$ and $\mathcal{X}_c^-$, while the PC value is added to the fitness set $\mathcal{F}^t$. Since the prompt is optimized with a triplet of images, the fitness value can also be assigned to the corresponding triplet in $\mathcal{S}^t$.

## 4.4 PROMPT REPRODUCTION

To generate diverse prompts that capture a broad range of class-specific features recognized by the victim model, we evolve the image triplet set $\mathcal{S}^t$ with $\mathcal{X}_c^s$, $\mathcal{X}_c^+$, and $\mathcal{X}_c^-$ as the candidate set. The triplet with the highest fitness value (PC) in $\mathcal{S}^t$ is selected as the elite, carried forward to the next generation $\mathcal{S}^{t+1}$ to guide the production of improved triplets. To generate new triplets, $N_p$ triplets are selected from $\mathcal{S}^t$, where $N_p$ denotes the number of parents. This is done by repeatedly sampling $k$ triplets and selecting the one with the highest fitness to form the parent set $\mathcal{S}_p$, a process known as tournament selection (Zames, 1981), where $k$ is the tournament size. Once the parent set is formed, two parent triplets are selected, and their images are randomly exchanged to create a new triplet, ensuring contributions from both parents. To introduce diversity, each image in the new triplet is

---

**Algorithm 1** Stealix

---

1: **Input:** target class $c$, seed image set $\mathcal{X}_c^s$, synthetic images amount $M$ for PC calculation, total query budget $B$, mutation probability $p_m$, population size $N$, parent size $N_p$, tournament size $k$, victim model $V$, generative model $G$, image encoder $I$ and text encoder $T$
2: **Output:** Attacker model $A$
3: Initialize attacker model $A$, $\mathcal{X}_c^+ \leftarrow \emptyset$, $\mathcal{X}_c^- \leftarrow \emptyset$, population index $t \leftarrow 0$, consumed budget $b \leftarrow 0$
4: // Initial population does not include samples from empty $\mathcal{X}_c^+, \mathcal{X}_c^-$
5: Initialize population $\mathcal{S}^t \leftarrow \{(\boldsymbol{x}_c^s, \boldsymbol{x}_c^+, \boldsymbol{x}_c^-)_i^t\}_{i=1}^N$ from $\mathcal{X}_c^s, \mathcal{X}_c^+, \mathcal{X}_c^-$
6: **while** $b < B$ **do**
7:    Initialize the fitness score set $\mathcal{F}^t \leftarrow \emptyset$
8:    **for** each triplet $(\boldsymbol{x}_c^s, \boldsymbol{x}_c^+, \boldsymbol{x}_c^-)_i^t$ in $\mathcal{S}^t$ **do**
9:       **if** $b \geq B$ **then**
10:          break
11:       **end if**
12:       // Optimize the randomly initialized prompt associated with the triplet (Section 4.2)
13:       $\boldsymbol{p}_i^t \leftarrow$ PromptRefinement$((\boldsymbol{x}_c^s, \boldsymbol{x}_c^+, \boldsymbol{x}_c^-)_i^t, I, T)$
14:       // Synthesize new images and calculate the prompt consistency as fitness (Section 4.3)
15:       $\{\boldsymbol{x}_i\}_{i=1}^M \sim$ G$(\boldsymbol{p}_i^t)$
16:       $\mathcal{F}^t \leftarrow \mathcal{F}^t \cup \{\frac{1}{M}\sum_{i=1}^M \mathbb{I}(V(\boldsymbol{x}_i) = c)\}$
17:       $b \leftarrow b + M$
18:       // Update the positive and negative sets
19:       $\mathcal{X}_c^+ \leftarrow \mathcal{X}_c^+ \cup \{\boldsymbol{x}_i \mid V(\boldsymbol{x}_i) = c, \ i \in \{1, \ldots, M\}\}$
20:       $\mathcal{X}_c^- \leftarrow \mathcal{X}_c^- \cup \{\boldsymbol{x}_i \mid V(\boldsymbol{x}_i) \neq c, \ i \in \{1, \ldots, M\}\}$
21:    **end for**
22:    // Apply prompt reproduction to generate the next population based on the fitness (Section 4.4).
23:    $\mathcal{S}^{t+1} \leftarrow$ PromptReproduction$(\mathcal{S}^t, \mathcal{F}^t, p_m, N_p, k, \mathcal{X}_c^s, \mathcal{X}_c^+, \mathcal{X}_c^-)$
24:    $t \leftarrow t + 1$
25: **end while**
26: Train model $A$ on images and victim labels using $\mathcal{X}_c^+, \mathcal{X}_c^-, \mathcal{X}_c^s$
27: **return** Attacker model $A$

---

replaced with a random sample from $\mathcal{X}_c^s$, $\mathcal{X}_c^+$, or $\mathcal{X}_c^-$ with a probability $p_m$, encouraging exploration of the candidate sets. The newly generated triplet is added to $\mathcal{S}^{t+1}$, and this process is repeated until the population is fully updated, balancing the preservation of high-fitness triplets with the generation of diverse new ones for refinement. See Algorithm 3 in Appendix A for more details of the prompt reproduction step.

## 5 EXPERIMENTS

### 5.1 EXPERIMENTAL SETUP

**Dataset.** We train the victim model on four datasets: EuroSAT (Helber et al., 2019), PASCAL VOC (Everingham et al., 2010), DomainNet (Peng et al., 2019), and CIFAR10 (Alex, 2009). Each dataset is chosen for its specific challenges in evaluating model stealing attacks. EuroSAT requires specialized prompts for satellite-based land use classification, as class names alone fail to generate relevant images. In PASCAL VOC, images are labeled by the largest object, testing the ability to identify the primary target from the victim model's prediction. DomainNet evaluates transfer learning across six diverse domains: clipart, infograph, paintings, quickdraw, real images, and sketches. A seed image is randomly chosen from one domain to test cross-domain generalization, using 10 of 345 classes for manageability. In CIFAR10, class names can guide image synthesis, leading to strong baselines when used by other methods, compared to ours, which does not. See Appendix B fore more details. In Appendix J, we introduce results on two medical datasets, highlighting the challenges when the diffusion model has limited domain-specific knowledge.

**Victim model.** All models use ResNet-34 following Truong et al. (2021), with PASCAL using an ImageNet-pretrained weights. Victim models are trained with SGD, Nesterov with momentum 0.9, a 0.01 learning rate, $5 \times 10^{-4}$ weight decay, and cosine annealing for 50 epochs.

**Stealix.** We use OpenCLIP-ViT/H as the vision-language model (Cherti et al., 2023) for prompt refinement, with a learning rate of 0.1 and 500 optimization steps using the AdamW optimizer. We employ Stable Diffusion-v2 (Rombach et al., 2022) as the generative model, with a guidance scale of 9 and 25 inference steps. PC evaluation uses $M = 10$ images. Stable Diffusion-v2 is used

across all methods. In prompt reproduction, we set the population size to $N = 10$, with $N_p = 5$ parents selected via tournament selection with a tournament size of $k = 5$, and retain one elite per generation. The mutation probability is set to $p_m = 0.6$. Following prior work (Truong et al., 2021), we use ResNet-18 as the attacker model. To focus on the impact of query data quality and ensure a fair comparison across methods, we train the attacker model using the same hyperparameters as the victim model without tuning: 50 epochs with SGD. More attacker and victim architecture setups are demonstrated in Appendix D and Appendix E. We run the experiments with a NVIDIA V100 32GB GPU and AMD EPYC 7543 32-Core Processor. The computation time is provided in Appendix I.

**Baselines.** We primarily focus on a different, more practical threat model that has not been explored before, where both prompt expertise and detailed class information are lacking. However, we also compare our method to existing approaches designed for other threat models. The results show that, even under this more challenging scenario, our method consistently outperforms the existing baselines. Specifically, we consider the following baselines. (i) **DA-Fusion** (Trabucco et al., 2024) is adapted to train a soft prompt from the seed image using textual inversion, then synthesize query images with strength 1 and the same guidance scale as our method; (ii) **Real Guidance** (He et al., 2023) uses the prompt "a photo of a {class name}" to synthesize images given the seed image with strength 1 and same guidance scale; (iii) **ASPKD** (Hondru & Ionescu, 2023) follows a three-stage process, first generating 5000 images per class using Real Guidance, then querying the victim model with a limited budget $B$, and finally pseudo-labeling the remaining images with a nearest neighbors approach with the attacker model; (iv) **Knockoff Nets** (Orekondy et al., 2019) evaluates performance with randomly collected images by querying the CIFAR-10 victim model with EuroSAT images and other victim models with CIFAR-10; (v) **DFME** (Truong et al., 2021) is a data-free model stealing method based on GANs that train a generator from scratch to adversarially generate samples to query the victim model. We report the result of DFME using a query budget of 2 million per class. (vi) **KD** (Hinton et al., 2015) serves as a reference upper bound, where the attacker queries the victim model using its training data to evaluate the best possible performance with data access. While data augmentation without querying the victim model is not model stealing, we include a comparison of attacker model accuracy between model stealing and data augmentation setups in Appendix H.

**Evaluation metrics.** We use two metrics for evaluation: (i) the accuracy of the attacker model on the test set of the victim data, which is a common evaluation protocol for model stealing attack (Orekondy et al., 2019) and (ii) the prompt consistency (PC) of the synthesized images. For Stealix, we report the best PC achieved across varying query budgets. For Real Guidance and DA-Fusion, where the prompt remains fixed, PC is measured by querying 500 images per class. For ASPKD that uses images synthesized by Real Guidance, PC is identical to Real Guidance. PC is not applicable for KD, Knockoff, and DFME, which do not involve text-to-image synthesis. All experiments are conducted with three random seeds, with mean values in the table and confidence intervals in the figure.

## 5.2 EXPERIMENTAL RESULTS

**Comparison with baselines.** Table 1 demonstrates a comparison of attacker model accuracy across methods, using a query budget of 500 per class (2M per class for DFME). Stealix consistently outperforms all other methods. For example, in CIFAR-10, Stealix achieves 49.6% accuracy, a 22.2% improvement over the second-best method, Real Guidance, which reaches 27.4%. DFME, by contrast, achieves near-random accuracy on EuroSAT and PASCAL due to its reliance on training a generator from scratch with small perturbations, which are quantized when interacting with real-world victim APIs, as discussed in Appendix G. In PASCAL, Stealix even surpasses KD, where the attacker has access to the victim data. This is because KD is constrained by the limited victim data size (around 73 images per class), whereas Stealix generates additional images and issues more queries, leading to better performance. In Figure 3 we illustrate both the attacker model accuracy and PC across varying query budgets. Stealix consistently achieves higher PC as the query budget increases, particularly in EuroSAT, where class names alone are insufficient for generating relevant images. Although Real Guidance initially attains higher PC in PASCAL and DomainNet, Stealix ultimately surpasses it with larger query budgets. In CIFAR-10, Stealix reaches nearly 100% PC. We also show in Appendix D and Appendix E that our method consistently outperforms others with different attacker model architectures and is resilient to variations in victim architectures.

Table 1: Attacker model accuracy with a query budget of 500 per class; DFME uses 2M.

| Method | #Seed images | Class name | EuroSAT | PASCAL | CIFAR10 | DomainNet |
|---|---|---|---|---|---|---|
| Victim | - | - | 98.2% (1.00x) | 82.7% (1.00x) | 93.8% (1.00x) | 83.9% (1.00x) |
| KD | - | - | 95.6% (0.97x) | 34.6% (0.42x) | 76.7% (0.82x) | 74.6% (0.89x) |
| Knockoff | 0 | ✗ | 40.1% (0.41x) | 22.3% (0.27x) | 24.4% (0.26x) | 39.3% (0.47x) |
| DFME | 0 | ✗ | 11.1% (0.11x) | 6.6% (0.08x) | 23.7% (0.25x) | 18.5% (0.22x) |
| ASPKD | 0 | ✓ | 39.2% (0.40x) | 25.7% (0.31x) | 27.1% (0.29x) | 27.3% (0.32x) |
| Real Guidance | 1 | ✓ | 51.2% (0.52x) | 24.0% (0.29x) | 27.4% (0.29x) | 31.9% (0.38x) |
| DA-Fusion | 1 | ✗ | 59.0% (0.60x) | 16.4% (0.20x) | 26.7% (0.28x) | 28.4% (0.34x) |
| Stealix (ours) | 1 | ✗ | **65.9%** (0.67x) | **40.0%** (0.48x) | **49.6%** (0.53x) | **48.4%** (0.58x) |

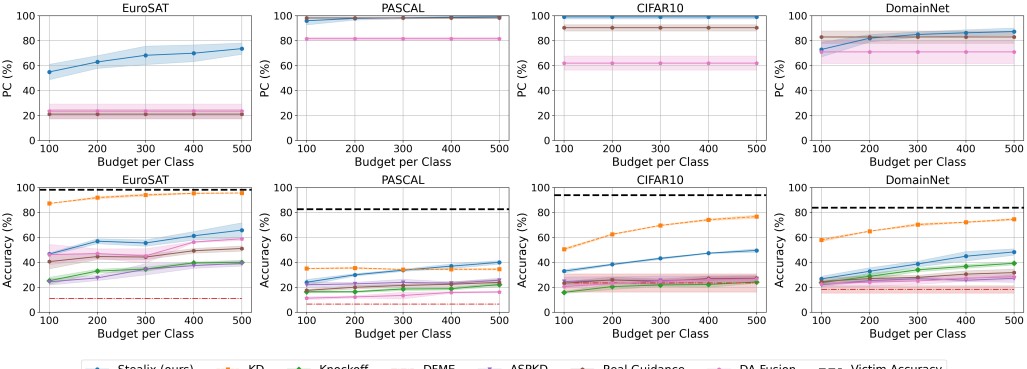

Figure 3: Attacker model accuracy (bottom) and PC (top) comparison across datasets with varying query budgets per class. DFME uses 2M per class.

**Limitations of human-crafted prompts.** Even when attackers can craft prompts for the seed image based on the prior knowledge of class names, these prompts, though logically accurate from a human perspective, may still fail to capture the nuanced features learned by the victim model. To evaluate this, we utilize InstructBLIP (Dai et al., 2023), a pre-trained vision-language model, as a proxy for a human attacker. InstructBLIP is instructed with, "It is a photo of a {class name}. Give me a prompt to synthesize similar images," alongside the seed image from the challenging EuroSAT dataset. The generated prompts for all classes are detailed in Appendix C. We synthesize 500 images per class based on these prompts and train the attacker model. The resulting PC and accuracy are shown in Table 2. Stealix outperforms InstructBLIP, achieving an accuracy of 65.9% compared to 55.2%. Despite InstructBLIP incorporating relevant terms like "aerial view" and "satellite view," its average PC score is 41.0%, compared to Stealix's 73.7%. For example, in the "Residential" class of EuroSAT (Figure 4), InstructBLIP's prompt "an aerial view of a residential area" results in a PC of only 8.8%, while Stealix reaches 71.0%. These findings emphasize the importance of prompt evolution in improving attacker model performance.

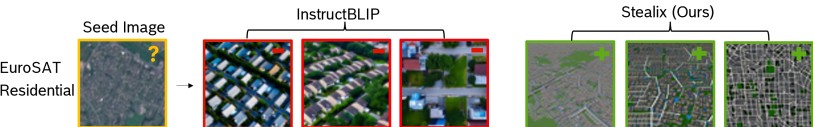

Figure 4: Synthetic images for the Residential class with the prompt from InstructBLIP and ours.

**Qualitative comparison.** Figure 5 presents qualitative comparisons on EuroSAT and PASCAL datasets. In EuroSAT, class names alone miss attributes like the satellite view, leading Real Guidance to generate generic images that differ from the victim data. Additionally, DA-Fusion struggles to interpret blurred seed images, generating random color blocks. For PASCAL, when multiple objects are present in the seed image, Stealix successfully identifies the target object. For example, ours removes the dog from the person class and correctly identifying the dining table as the target instead of the human beside it, while DA-Fusion mistakenly targets the wrong objects.

**Correlation between PC and feature distance.** Since the attacker lacks access to the distribution of the victim data, PC is proposed as a proxy for monitoring and optimizing prompts, based on the

Table 2: Comparison with InstructBLIP on EuroSAT at a query budget of 500 per class.

| Method | #Seed images | Class name | PC | Accuracy |
|---|---|---|---|---|
| InstructBLIP | 1 | ✓ | 41.0% | 55.2% |
| Stealix (ours) | 1 | ✗ | **73.7%** | **65.9%** |

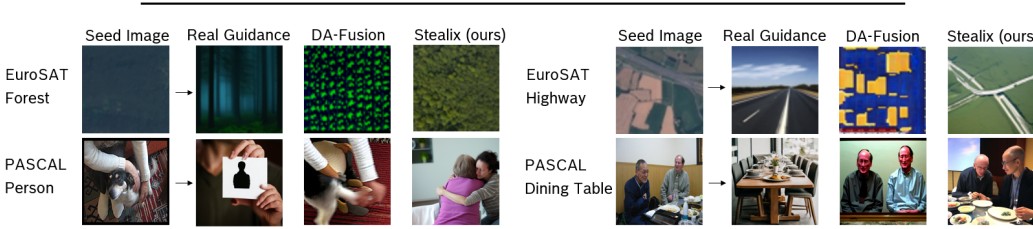

Figure 5: Qualitative comparison of images generated by Real Guidance, DA-Fusion, and Stealix on the EuroSAT and PASCAL datasets. Other baselines include: Knockoff uses CIFAR10 as query data, DFME synthesizes noise images, and ASPKD uses the same images as Real Guidance.

hypothesis that more consistent predictions from the victim model indicate a closer match to its data. To evaluate this assumption, we collect 150 PC values per class corresponding to different prompts during prompt evolution. For each PC, we compute the mean feature vector of the synthetic images and calculate its $L_2$ distance from the mean feature vector of the victim data. Feature vectors are extracted from the victim model before its final fully connected layer. Spearman's rank correlation test shows a strong, statistically significant negative correlation between PC and $L_2$, as summarized in Table 3, supporting our assumption.

**Linking prompt consistency to model accuracy.** To evaluate whether higher PC leads to more effective model stealing, we compare attacker model performance using synthetic images generated from two prompts with different PC values. Specifically, we select prompts at the 25th percentile (lower PC) and the 100th percentile (higher PC) during the prompt evolution process. We generate 500 synthetic images with each of the two prompts, query the victim model, and use only the positive images to train the attacker model. We exclude the 0th percentile prompt because it yields no positive samples. Since the higher PC prompt generates more positive images than the lower PC prompt, we reduce the number of positive images from the higher PC prompt to match that of the lower PC prompt. The results, presented in Figure 6, demonstrate that higher PC values consistently lead to improved attacker model accuracy across all datasets, confirming that higher PC enhances the effectiveness of model stealing attacks.

**Diversity comparison.** Figure 3 shows that although PC values of Real Guidance are similar to ours for PASCAL and DomainNet, our attacker model performs consistently better. This advantage stems from the greater diversity in our synthetic data, achieved through prompt evolution, where distinct images are used to construct different triplets. To quantify this, we use the diversity score proposed by Kynkäänniemi et al. (2019), Recall, which measures the likelihood that a random image from the victim data distribution falls within the support of the synthetic image set. The higher the score, the more diverse the images. As shown in Table 4, our method generates more diverse synthetic data with higher Recall score.

**Ablative analysis.** We evaluate the effectiveness of prompt reproduction by conducting an ablation study, where prompts are optimized using only CLIP from the seed image, without reproduction. As shown in Table 5, labeled "Stealix w/o reproduction", the accuracy drops significantly, highlighting the critical role of the reproduction to evolve the prompts with prompt consistency and refinement.

## 6 DISCUSSION

**Defense.** In our threat model, we assume the victim employs the defense of providing only hard-label outputs, which is effective at limiting information leakage compared to soft labels (Sanyal et al., 2022) without adding computational overhead to the victim's system. As demonstrated in Appendix F, the attacker model accuracy improves with soft-label access using images generated by Stealix, underscoring the need for this defense. However, since our prompt evolution method only relies on hard labels, it remains effective, suggesting more advanced defenses may be necessary.

Table 3: Spearman's rank correlation between PC and $L_2$ feature distance.

| Data | Correlation $\rho$ | p-value |
|---|---|---|
| EuroSAT | $-0.63$ | $7.04 \times 10^{-5}$ |
| PASCAL | $-0.64$ | $2.79 \times 10^{-4}$ |
| CIFAR10 | $-0.73$ | $1.20 \times 10^{-7}$ |
| DomainNet | $-0.88$ | $1.83 \times 10^{-26}$ |

Figure 6: Comparison of attacker model accuracy using synthetic images generated from prompts with higher and lower prompt consistency across four datasets.

Table 4: Diversity (recall) comparison across methods that using pre-trained text-to-image generative models, with higher scores indicating greater diversity relative to the victim data distribution.

| Method | EuroSAT | PASCAL | CIFAR10 | DomainNet |
|---|---|---|---|---|
| Real Guidance | 0.29 | 0.07 | 0.40 | 0.41 |
| DA-Fusion | 0.43 | 0.06 | 0.48 | 0.24 |
| Stealix (ours) | **0.49** | **0.30** | **0.76** | **0.66** |

Table 5: Ablation study: comparison of attacker model accuracy without prompt reproduction.

| Method | EuroSAT | PASCAL | CIFAR10 | DomainNet |
|---|---|---|---|---|
| Victim | 98.2% (1.00x) | 82.7% (1.00x) | 93.8% (1.00x) | 83.9% (1.00x) |
| Stealix w/o reproduction | 60.1% (0.61x) | 26.7% (0.32x) | 33.8% (0.36x) | 39.2% (0.47x) |
| Stealix (ours) | **65.9%** (0.67x) | **40.0%** (0.48x) | **49.6%** (0.53x) | **48.4%** (0.58x) |

**Limitations and future work.** Our approach, unlike GAN-based methods, does not require backpropagation through the victim model to train the generator, which enhances generalization across victim model architectures (Appendix E). Although the attacker model architecture can still influence the performance (Appendix D), our method consistently outperforms the baselines. Furthermore, since image synthesis and attacker model training are decoupled, attackers can reuse synthetic images for hyperparameter tuning and neural architecture search. This key advantage could be further explored in future work to improve model accuracy. Finally, as open-source generative models advance, integrating more powerful models into our framework offers significant potential for further enhancements.

# 7 CONCLUSION

We demonstrate that attackers can leverage open-source generative models to steal proprietary models, even without expertise in prompt design or access to class information. Without direct access to victim data, we show that prompt evolution as done by Stealix significantly improves model extraction efficiency. Furthermore, we underscore the crucial role of matching the similarity between the generated data with the victim data, which enhances the effectiveness of the attack. This is the first study to expose the risks posed by publicly available pre-trained generative models in model theft under a realistic attack setting. We call for more attention toward developing defense mechanisms to mitigate this emerging threat.

## REPRODUCIBILITY STATEMENT

The authors are committed to ensuring the reproducibility of this work. The appendix provides extensive implementation details, and the code and setup will be made publicly available as open-source.

ETHICS STATEMENT

This work aims to raise awareness of the risks associated with model stealing, particularly through the use of open-source pre-trained generative models. While our work demonstrates how such models can be exploited in adversarial settings, it is intended to inform the development of more robust defenses against model theft. We emphasize that our approach is not designed to promote malicious behavior but to highlight vulnerabilities that need addressing within the AI community. We encourage practitioners, model developers, and stakeholders to implement stronger defenses, such as hard-label-only responses or adversarial detection mechanisms, to mitigate potential risks. All experiments were conducted with publicly available models and data, and with the intent of advancing the security of AI systems.

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

## A  ALGORITHMS

We detail the algorithms for prompt refinement and prompt reproduction in Section 4.2 and Section 4.4.

**Prompt refinement.** We implement the hard prompt optimization method proposed by Wen et al. (2024) to optimize the prompt to capture target class features learnt by the victim model (Algorithm 2). The soft prompt, $\hat{\boldsymbol{p}}$, consists of $L$ embedding vectors and is initialized from the vocabulary embedding set $\mathbf{E}$. The soft prompt is iteratively mapped to its nearest neighbor embeddings using a projection function, $\text{Proj}\mathbf{E}(\hat{\boldsymbol{p}})$, and converted into a hard prompt, $\boldsymbol{p}$, via a function $\text{Soft2Hard}(\hat{\boldsymbol{p}})$. During each iteration, the soft prompt is updated through gradient descent, guided by the similarity loss $\mathcal{L}_{\text{SIM}}$, which aims to retain features in the positive image while reducing features in the negative image. This process is repeated for $s$ optimization steps, after which the final hard prompt is obtained. We follow the hyperparameters from Wen et al. (2024), setting $L = 16$ and $\gamma = 0.1$, while reducing $s$ from 5000 to 500 to save optimization time, e.g., on EuroSAT, from approximately 3 minutes to 18 seconds. We further evaluate the impact of prompt lengths (4, 16, 32) on EuroSAT with a query budget of 500 per class across three random seeds. Table 6 shows that Stealix consistently outperforms others (best baseline: 59.0% from DA-Fusion in Table 1), with prompt length 16 striking the best balance between efficiency and accuracy.

---

**Algorithm 2** Prompt Refinment

1: **Input:** image triplet $(\boldsymbol{x}_c^s, \boldsymbol{x}_c^+, \boldsymbol{x}_c^-)$, text encoder $T$ and image encoder $I$, optimization steps $s$, learning rate $\gamma$, soft prompt length $L$
2: **Output:** hard prompt $\boldsymbol{p}$
3: Initialize soft prompt $\hat{\boldsymbol{p}}$ from vocabulary $\mathbf{E}$
4: **for** step $= 1$ to $s$ **do**
5:     // Project soft prompt to nearest neighbor embeddings and convert to hard prompt.
6:     $\hat{\boldsymbol{p}}' \leftarrow \text{Proj}_{\mathbf{E}}(\hat{\boldsymbol{p}})$
7:     $\boldsymbol{p} \leftarrow \text{Soft2Hard}(\hat{\boldsymbol{p}}')$
8:     // Compute gradient of the similarity loss and update soft prompt using gradient descent.
9:     $g \leftarrow \nabla_{\hat{\boldsymbol{p}}'} \sum_{\mathbf{x} \in (\boldsymbol{x}_c^s, \boldsymbol{x}_c^+, \boldsymbol{x}_c^-)} \mathcal{L}_{\text{SIM}}(I(\boldsymbol{x}), T(\boldsymbol{p}), V(\boldsymbol{x}))$
10:    $\hat{\boldsymbol{p}} \leftarrow \hat{\boldsymbol{p}} - \gamma g$
11: **end for**
12: // Final projection to ensure the soft prompt is fully converted to hard tokens.
13: $\hat{\boldsymbol{p}}' \leftarrow \text{Proj}_{\mathbf{E}}(\hat{\boldsymbol{p}})$
14: $\boldsymbol{p} \leftarrow \text{Soft2Hard}(\hat{\boldsymbol{p}}')$
15: **return** hard prompt $\boldsymbol{p}$

---

Table 6: Attacker model accuracy with different prompt lengths on EuroSAT using Stealix. The victim model accuracy is 98.2% and the second best baseline is 59.0%.

| Prompt length | 4 | 16 | 32 |
|---|---|---|---|
| Stealix | 62.5% | 65.9% | 64.3% |

**Prompt reproduction.** In Algorithm 3, we employ a genetic algorithm to iteratively refine prompts through tournament selection, crossover and mutation. In tournament selection, we use prompt consistency as the fitness function.

## B  DATASETS

We provide an overview of the datasets introduced in our experiment setup (Section 5.1), detailing the sizes of the training and validation sets and their respective image resolutions (see Table 7). For CIFAR-10, we utilize the standard training and test splits provided by PyTorch, which consist of 50,000 training images and 10,000 test images at a resolution of $32 \times 32$ pixels. In the case of PASCAL, we follow the preprocess from DA-Fusion (Trabucco et al., 2024) to assign classification labels based on the largest object present in each image, resulting in 1,464 training images and

---

**Algorithm 3** Prompt Reproduction

---

1: **Input:** Current population $\mathcal{S}^t$, fitness set $\mathcal{F}^t$, seed image set $\mathcal{X}_c^s$, positive image set $\mathcal{X}_c^+$, negative image set $\mathcal{X}_c^-$, tournament size $k$, number of parents $N_p$, number of populations $N$.
2: **Output:** Evolved population $\mathcal{S}^{t+1}$
3: Select the elite triplet $(x_c^s, x_c^+, x_c^-)_{\text{elite}}$ with the highest fitness from $\mathcal{S}^t$ given $\mathcal{F}^t$
4: Initialize next population $\mathcal{S}^{t+1} \leftarrow \{(x_c^s, x_c^+, x_c^-)_{\text{elite}}\}$ // Keep the elite triplet in the next population
5: Initialize the parents set $\mathcal{S}_p \leftarrow \emptyset$
6: // Perform tournament selection to select $N_p$ parents.
7: **for** $i = 1$ to $N_p$ **do**
8:     Randomly select $k$ triplets from $\mathcal{S}^t$
9:     Choose the triplet $(x_c^s, x_c^+, x_c^-)$ with maximum fitness from the $k$ triplets given $\mathcal{F}^t$
10:     $\mathcal{S}_p \leftarrow \mathcal{S}_p \cup \{(x_c^s, x_c^+, x_c^-)\}$
11: **end for**
12: // Generate the next generation.
13: **for** $i = 1$ to $N - 1$ **do**
14:     // Apply crossover using selected parents.
15:     Select 2 parents from $\mathcal{S}_p$ cyclically
16:     Split each parent at a random point
17:     Form a new triplet $(x_c^s, x_c^+, x_c^-)$ by combining the left part of one parent with the right part of the other
18:     // Apply mutation.
19:     Replace each image in $(x_c^s, x_c^+, x_c^-)$ with a random one from $\mathcal{X}_c^s$, $\mathcal{X}_c^+$, or $\mathcal{X}_c^-$ with probability $p_m$
20:     $\mathcal{S}^{t+1} \leftarrow \mathcal{S}^{t+1} \cup \{(x_c^s, x_c^+, x_c^-)\}$
21: **end for**
22: $\mathcal{S}^{t+1}$

---

1,449 validation images with an image size of $256 \times 256$ pixels. The EuroSAT dataset is split into training and validation sets using an 80/20 ratio while maintaining class distribution through stratified sampling, yielding 21,600 training images and 5,400 validation images at a resolution of $64 \times 64$ pixels. For DomainNet, we select the first 10 classes in alphabetical order across six diverse domains: clipart, infograph, paintings, quickdraw, real images, and sketches. We apply the same 80/20 stratified split as used for EuroSAT, resulting in 11,449 training images and 2,863 validation images, each resized to $64 \times 64$ pixels.

Table 7: Overview of datasets.

| Dataset | Train/Val | Image Size |
|---------|-----------|------------|
| EuroSAT | 21.6K/5.4K | $64 \times 64$ |
| PASCAL | 1464/1449 | $256 \times 256$ |
| CIFAR10 | 50K/10K | $32 \times 32$ |
| DomainNet | 11449/2863 | $64 \times 64$ |

## C    SIMULATING ATTACKER WITH INSTRUCTBLIP

The prompts generated by InstructBLIP (Dai et al., 2023) for the EuroSAT dataset are based on the instruction: "It is a photo of a {class name}. Give me a prompt to synthesize similar images." The prompts for each class are listed in Table 8. Performance differences between these prompts and Stealix are discussed in Section 5.2.

## D    DIFFERENT ATTACKER MODEL ARCHITECTURES

We analyze the performance of different attacker model architectures, including ResNet18, ResNet34, VGG16, and MobileNet, as shown in Table 9. Our method, Stealix, consistently outperforms all other baselines, regardless of the attacker model architecture. However, the choice of architecture does impact performance: smaller models like MobileNet result in lower accuracy due to their limited capacity, as seen in the KD baseline where MobileNet achieves only 89.2% accuracy compared to 95.6% with ResNet. This suggests that architectural limitations, rather than the attack method, drive the performance drop. Moreover, because Stealix decouples image synthesis from attacker model training, attackers can optimize hyperparameters and architectures without re-querying the victim model, offering flexibility and efficiency.

Table 8: Generated prompts from InstructBLIP for EuroSAT classes.

| Class name | Generated prompt |
|---|---|
| AnnualCrop | "an aerial view of a farm in the countryside" |
| Forest | "an aerial view of a forest" |
| HerbaceousVegetation | "a satellite image of the earth taken from space" |
| Highway | "an aerial view of a highway and farmland" |
| Industrial | "an aerial view of a large industrial area" |
| Pasture | "an aerial view of a farm" |
| PermanentCrop | "an aerial view of a farm" |
| Residential | "an aerial view of a residential area" |
| River | "an aerial view of a river" |
| SeaLake | "an aerial view of a large body of water" |

Table 9: Performance comparison of different attacker architectures against a ResNet34 victim model (98.2% accuracy) trained on EuroSAT, using a query budget of 500 queries per class.

| Method | #Seed images | Class name | Attacker architecture | | | |
|---|---|---|---|---|---|---|
| | | | ResNet18 | ResNet34 | VGG16 | MobileNet |
| KD | - | - | 95.6% (0.97x) | 95.6% (0.97x) | 95.7% (0.97x) | 89.2% (0.91x) |
| Knockoff | 0 | ✗ | 40.1% (0.41x) | 40.3% (0.41x) | 40.1% (0.41x) | 29.3% (0.30x) |
| DFME | 0 | ✗ | 11.1% (0.11x) | 11.1% (0.11x) | 11.1% (0.11x) | 11.1% (0.11x) |
| ASPKD | 0 | ✓ | 39.2% (0.40x) | 39.0% (0.40x) | 35.4% (0.36x) | 32.0% (0.33x) |
| Real Guidance | 1 | ✓ | 51.2% (0.52x) | 52.0% (0.53x) | 43.9% (0.45x) | 40.6% (0.41x) |
| DA-Fusion | 1 | ✗ | 59.0% (0.60x) | 53.3% (0.54x) | 58.8% (0.60x) | 48.6% (0.50x) |
| Stealix (ours) | 1 | ✗ | **65.9%** (0.67x) | **67.9%** (0.69x) | **66.0%** (0.67x) | **51.9%** (0.53x) |

# E    DIFFERENT VICTIM MODEL ARCHITECTURES

We analyze the performance of Stealix across different victim model architectures on EuroSAT, including ResNet18, ResNet34, VGG16, and MobileNet, as shown in Table 10. Using ResNet18 as the attacker architecture, Stealix consistently performs well across these architectures, demonstrating its robustness to variations in the victim model. The ability to generalize across diverse architectures highlights the adaptability and effectiveness of Stealix in real-world scenarios where the attacker may not know the exact architecture of the victim model.

Table 10: Performance comparison of Stealix against different victim architectures (ResNet18, ResNet34, VGG16, MobileNet) with the attacker model architecture set to ResNet18 across all experiments on EuroSAT.

| Method | Victim architecture | | | |
|---|---|---|---|---|
| | ResNet18 | ResNet34 | VGG16 | MobileNet |
| Victim | 98.4% (1.00x) | 98.2% (1.00x) | 98.2% (1.00x) | 96.9% (1.00x) |
| Stealix (ResNet18) | 66.2% (0.67x) | 65.9% (0.67x) | 73.4% (0.75x) | 66.0% (0.68x) |

# F    STEALIX WITH SOFT LABELS

In this experiment, we evaluate the impact of soft-label access on the attacker model accuracy compared to the hard-label-only scenario. Since Stealix's prompt evolution only relies on hard labels for calculating prompt consistency, the same synthetic images are used to train the attacker model under both conditions, with the only difference being whether the labels are hard or soft (full probability predictions). As shown in Figure 7, Stealix consistently achieves higher accuracy with soft-label access across all datasets, as soft labels provide richer information through confidence scores, resulting in improved model performance. This underscores the importance of defenses like hard-label-only outputs to limit the effectiveness of model stealing attacks. However, hard-label defenses merely slow down the attack, increasing the required query budget without fully preventing model theft. Given the high quality and alignment of synthetic images with the victim's data, the attack remains

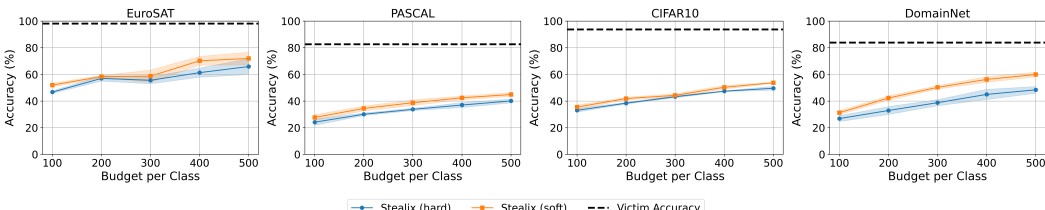

Figure 7: Performance comparison of Stealix with hard label and soft label access across EuroSAT, PASCAL, CIFAR-10, and DomainNet at varying query budgets.

viable over time. This highlights the need for more advanced defense strategies to better address this threat in future research.

# G  LIMITATIONS OF DFME

We analyze the performance of DFME (Truong et al., 2021) under realistic attack scenarios. Following the original DFME setup, we attempted to extract our ResNet34 victim model trained on CIFAR-10 using 2 million queries per class with soft-label access, achieving an attacker model accuracy of 87.4%, which is comparable to the results reported in the original work. However, DFME generates images with pixel values in the range $(-1, 1)$ due to the use of Tanh activation, which is incompatible with real-world APIs that expect standard image formats (e.g., pixel values in $[0, 255]$). After quantizing these images to the standard format, the attacker model accuracy dropped to 76.4%, despite using the same query budget. This performance degradation occurs because DFME relies on adding small perturbations to the generated images to estimate gradients via forward differences (Wibisono et al., 2012). Quantization can negate these subtle perturbations. Furthermore, when the victim model provides only hard-label outputs as a defense mechanism, the attacker model accuracy further decreased to 23.7%. In this case, the output labels remain constant under small input perturbations, rendering forward difference methods ineffective for gradient estimation and significantly limiting the attacker's ability to train the generator.

We present the results across all datasets in Table 11. In the case of PASCAL, we reduced the batch size from 256 to 64 due to computational constraints imposed by the large image size ($256 \times 256$ pixels). Notably, DFME fails to extract the PASCAL victim model, likely due to this higher image resolution. Furthermore, for the fine-grained EuroSAT dataset, even with soft-label access and without quantization, the attacker model achieves only 19.0% accuracy.

Table 11: Performance of DFME on various datasets under different settings with a query budget of 2M per class. Victim model accuracies are provided for reference.

| Method | EuroSAT | PASCAL | CIFAR10 | DomainNet |
|---|---|---|---|---|
| Victim | 98.2% (1.00x) | 82.7% (1.00x) | 93.8% (1.00x) | 83.9% (1.00x) |
| DFME | 19.0% (0.19x) | 6.6% (0.08x) | 87.4% (0.93x) | 83.0% (0.99x) |
| + Quantization | 10.2% (0.10x) | 6.6% (0.08x) | 76.4% (0.81x) | 72.0% (0.86x) |
| + Hard label | 11.1% (0.11x) | 6.6% (0.08x) | 23.7% (0.25x) | 18.5% (0.22x) |

## H  DA-FUSION AS DATA AUGMENTATION

Having one image per class is a realistic setup and differs from having full access to victim data or its distribution. This reflects real-world threats posed by competitors in the same field, aiming to provide similar services. While attackers can use DA-Fusion to augment the seed images to train the attacker model without querying the victim model, we demonstrate that model stealing still provides a substantial performance improvement. We compare the accuracy of attacker models under a model stealing setup versus a data augmentation setup, with a query budget of 500 per class. Table 12 shows that performance degrades significantly with DA-Fusion when relying solely on class labels for training instead of using predictions from the victim model, highlighting that model stealing is essential, even with one image per class.

Table 12: Comparison of attacker model training with and without victim queries, showing accuracy with a 500-query budget per class; DFME uses 2M.

| Method | Query victim | EuroSAT | PASCAL | CIFAR10 | DomainNet |
|---|---|---|---|---|---|
| Victim | - | 98.2% (1.00x) | 82.7% (1.00x) | 93.8% (1.00x) | 83.9% (1.00x) |
| Stealix (ours) | ✓ | **65.9%** (0.67x) | **40.0%** (0.48x) | **49.6%** (0.53x) | **48.4%** (0.58x) |
| DA-Fusion | ✓ | 59.0% (0.60x) | 16.4% (0.20x) | 26.7% (0.28x) | 28.4% (0.34x) |
| DA-Fusion | ✗ | 29.9% (0.30x) | 10.7% (0.13x) | 18.9% (0.20x) | 17.9% (0.21x) |

## I  COMPARISON OF COMPUTATION TIME

We present a comparison of the time required for various methods using the EuroSAT dataset as an example. All experiments were conducted on a single machine with an NVIDIA V100 32GB GPU and an AMD EPYC 7543 32-Core Processor. Table 13 summarizes the total time for the process under a 500-query budget per class (with DFME using 2M queries per class). Stealix demonstrates state-of-the-art accuracy while maintaining reasonable computational efficiency.

Table 13: Comparison of computational time and accuracy across methods on the EuroSAT dataset. The victim model accuracy 98.2%.

| | Knockoff | DFME | ASPKD | Real Guidance | DA-Fusion | Stealix (ours) |
|---|---|---|---|---|---|---|
| **Time (hours)** | 0.5 | 4.5 | 28.6 | 3.3 | 5.4 | 6.3 |
| **Accuracy** | 40.1% | 11.1% | 39.2% | 51.2% | 59.0% | 65.9% |

## J  LIMITED MEDICAL KNOWLEDGE

As generative priors like diffusion models are trained on public available data, the absence or limited presence of domain-specific knowledge, such as medical expertise, would have impact on the performance of model stealing. However, this issue applies universally to all model stealing methods that rely on diffusion models, not specifically to ours. Our experiment results in Table 1 show that diffusion models can be leveraged more effectively in model stealing when they describe the data well but are not properly prompted. In other words, **our approach shares the same lower-bound as existing methods but significantly improves the upper-bound**, achieving an approximate 7–22% improvement compared to the second-best method, as shown in Table 1.

With that being said, we conducted an experiment analyzing performance when diffusion models have limited domain-specific knowledge. We consider two medical datasets: PatchCamelyon (PCAM) (Veeling et al., 2018) and RetinaMNIST (Yang et al., 2023). In PCAM, class names are 'benign tissue' and 'tumor tissue'. RetinaMNIST involves a 5-level grading system for diabetic retinopathy severity, with class names as 'diabetic retinopathy $i$,' where $i$ ranges from 0 to 4 for severity. We conduct experiments using three random seeds and report the mean attacker accuracy below, following the setup described in Section 5.1. The victim model uses the ResNet34 architecture, while the attacker model is based on ResNet18. The qualitative comparison in Figure 8 shows

Table 14: Attacker model accuracy for medical dataset with a query budget of 500 per class; DFME uses 2M.

| Method | #Seed images | Class name | PCAM | RetinaMNIST |
|---|---|---|---|---|
| Victim | - | - | 91.2% (1.00x) | 61.7% (1.00x) |
| KD | - | - | 76.3% (0.84x) | 59.4% (0.96x) |
| Knockoff | 0 | ✗ | 50.0% (0.55x) | 56.1% (0.91x) |
| DFME | 0 | ✗ | 50.0% (0.55x) | 46.1% (0.75x) |
| ASPKD | 0 | ✓ | 60.1% (0.66x) | 55.3% (0.90x) |
| Real Guidance | 1 | ✓ | 61.8% (0.68x) | 56.1% (0.91x) |
| DA-Fusion | 1 | ✗ | 61.5% (0.68x) | 56.7% (0.92x) |
| Stealix (ours) | 1 | ✗ | **62.2%** (0.68x) | **58.0%** (0.94x) |

that the diffusion model struggles to synthesize Retina-like images, highlighting its limited knowledge. However, the results in Table 14 show that methods with generative priors still outperform Knockoff and DFME, affirming the value of priors, though the improvements decrease as the data deviates from diffusion model's distribution, resulting in only modest gains of Stealix in such cases.

In summary, our approach provides (1) significant improvement when diffusion models can describe the data and (2) comparable or slightly better performance when they have limited domain knowledge.

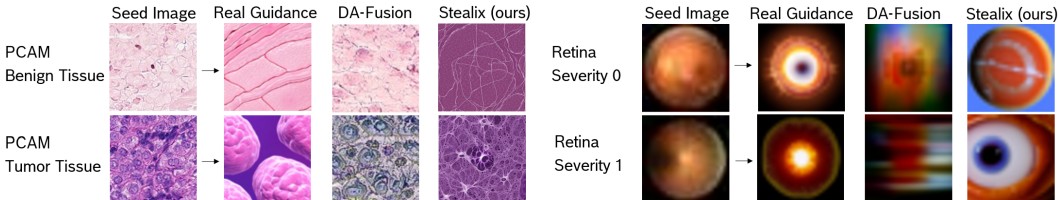

Figure 8: Qualitative comparison of images generated by Real Guidance, DA-Fusion, and Stealix on the PCAM and RetinaMNIST datasets. Other baselines include: Knockoff uses CIFAR10 as query data, DFME synthesizes noise images, and ASPKD uses the same images as Real Guidance.

