# OpenReview forum: "Stealix: Model Stealing via Prompt Evolution"
_ICLR.cc/2025/Conference — Submitted to ICLR 2025_

### Official Review · Reviewer_9Eda · 2024-10-19

**Soundness:** 3
**Presentation:** 3
**Contribution:** 3
**Rating:** 6
**Confidence:** 3

**Summary:**

The paper introduces a novel method for model stealing by leveraging off-the-shelf diffusion models to synthesize query images. More specifically, the proposed method uses prompt optimization to automatically craft reasonable prompts and genetic optimization to increase the diversity of the prompts and their corresponding image generations. Prompt optimization is done using a hard prompt optimization by maximizing the CLIP similarity between the generated image and the predicted class by the target model (if the prediction is correct) or minimizing the similarity if the target model's prediction is wrong. The genetic optimization relies on image triplets consisting of a base sample of the target class, a positive, correctly classified synthetic image, and an incorrectly classified image. By iteratively optimizing these image triplets using prompt consistency as a metric, the method creates a diverse set of images that are then used to train the surrogate model that imitates the victim model.

**Strengths:**

- The paper is clearly written and easy to follow. Notably, all hyperparameters and settings are explicitly described and stated, supporting reproducibility. I enjoyed reading this paper.
- Using genetic optimization in combination with prompt optimization to craft a diverse set of attack samples is a clever attack method compared to hand-crafted prompts. Removing the human factor from the pipeline speeds up the process and increases the parallelization of the attack.
- The results are convincing, clearly beating existing baselines by a significant margin. Furthermore, multiple ablation and sensitivity studies are conducted, including multiple datasets, architectures, correlation analyses, and an image diversity analysis. Each of these analyses offers interesting insights.

**Weaknesses:**

- The method primarily relies on existing work on genetic optimization and hard prompt optimization. While combining these methods is novel in the domain of model stealing, the individual parts already exist in literature. I do not find this compellingly flawed but I want to mention that the paper proposes no novel, technical contributions besides combining existing methods.
- The method overview in 4.1 does not seem exact. Why should the attack be interested in minimizing the cross-entropy loss between the victim model and the surrogate model under label-only access? From my understanding of the paper, the attack goal is to replicate the target model's label prediction performance and not to imitate the distribution of the confidence scores (which the cross-entropy loss measures). I think a correct target would be to align the label predictions, i.e., the predicted labels by the victim and surrogate models should match, not their prediction score distribution, which also aligns with the metrics used in Table 1.
- The experiments seem to be conducted only once per setup. For stable results, each experiment should be repeated at least three times with different seeds to eliminate influences due to randomness in the training/attack process.

Small Remarks:
- L144: Using the term "label-only access" might be a more explicit description of the setting compared to "black box."
- The attack goal should be formulated in a more precise way. From the description in L150ff, it is not exactly clear what the attack goal is. Should the surrogate model "steal" the target model's performance in the sense that it should maximize its accuracy on a test set? Or does the stealing attack aim to extract the target model's parameters, trying to replicate the model? It becomes clearer later in the paper, but a more precise definition early in the paper would be helpful for understanding.

**Questions:**

- Is there an explanation why the knowledge distillation accuracy in Table 1 is, in some cases, substantially lower than the victim model's test accuracy? I understand that vanilla knowledge distillation has some drawbacks, but I expect the student model's accuracy to be better, for example, in the CIFAR10 case.
- How are the qualitative samples from Fig. 5 picked? Are they cherry-picked or randomly selected?

---

> ### Author Response · Authors · 2024-11-20
> **Response to Reviewer 9Eda**
>
> We thank the reviewer for their valuable suggestions. We are glad they enjoy reading our paper, find the method novel, and think the insights in our analysis interesting. We would like to address the following questions.
>
> > W1. The method primarily relies on existing work on genetic optimization and hard prompt optimization. While combining these methods is novel in the domain of model stealing, the individual parts already exist in literature. I do not find this compellingly flawed but I want to mention that the paper proposes no novel, technical contributions besides combining existing methods.
>
> We appreciate the acknowledgment of our method as a novel contribution to model stealing. While we use genetic and hard prompt optimization, each component of our method addresses a unique challenge unexplored in previous works, as discussed in the general response [G1](https://openreview.net/forum?id=kvN8MJTOCM&noteId=dJYaD8RKq6).
>
>
>
> > W2. Why should the attack be interested in minimizing the cross-entropy loss between the victim model and the surrogate model under label-only access?
>
>
> We use the cross-entropy loss as a proxy for the predicted label (0-1 loss), in the same way that this former is traditionally used for training classification models as a differentiable approximation to counting model errors (0-1 loss here as well). We do not find this practice misaligned with using accuracy as a performance metric (Tab. 1). It is also a standard practice, including in the model theft community [1, 2]. Although alternative criteria may be worth exploring, such investigations are beyond the scope of this paper.
>
>
> > W3. The experiments seem to be conducted only once per setup.
>
> We did conduct the experiments with **three different random seeds** (see confidence intervals in Fig. 3 and 6). All accuracy and corresponding PC values in the tables represent the mean across the three repetitions. We will update the paper to clearly state the number of experiment repeatitions.
>
> > W4. L144: Using the term "label-only access" might be a more explicit description of the setting compared to "black box."
>
> Thanks for the suggestion. We will update the terminology in the paper for clarity.
>
> > W5. ... (The attack goal) becomes clearer later in the paper, but a more precise definition early in the paper would be helpful for understanding.
>
> The attack goal is to "maximize its accuracy on a test set" as the reviewer mentioned. We will give this definition earlier.
>
> > Q1. Is there an explanation why the knowledge distillation accuracy in Table 1 is, in some cases, substantially lower than the victim model's test accuracy?
>
> For the knowledge distillation (KD) setting in Tab. 1, the attacker has only 500 queries per class and hard label access, rather than the entire victim dataset. As shown in Fig. 3, performance improves with a larger budget. For PASCAL, the gap arises because the victim model uses ImageNet-pretrained weights for ResNet34, while the attacker does not.
>
> > Q2. How are the qualitative samples from Fig. 5 picked? Are they cherry-picked or randomly selected?
>
> For Stealix, samples are randomly picked from images generated by prompts with higher prompt consistency that we optimize. For others, they are randomly picked.
>
>
> [1] Tribhuvanesh Orekondy, Bernt Schiele, and Mario Fritz. Knockoff nets: Stealing functionality of
> black-box models. In Conference on Computer Vision and Pattern Recognition (CVPR), 2019
>
> [2] Vlad Hondru and Radu Tudor Ionescu. Towards few-call model stealing via active self-paced knowledge distillation and diffusion-based image generation. arXiv preprint arXiv:2310.00096, 2023.

---

> > ### Comment · Reviewer_9Eda · 2024-11-21
> >
> > I thank the authors for the clarification of my questions and remarks (and all the other reviewers' questions). I just have one question left regarding their answer to Q1:
> >
> > > For PASCAL, the gap arises because the victim model uses ImageNet-pretrained weights for ResNet34, while the attacker does not.
> >
> > Would it not make sense to also evaluate and compare the approaches when using ImageNet initializations? In practice, almost everyone trains their models on some pre-trained weights like ImageNet, so in my opinion, a realistic setting should also include pre-trained weights.

---

> > > ### Author Response · Authors · 2024-11-21
> > > **Initialize with pretrained weights**
> > >
> > > We initialized all victim and attacker models without any pretrained weights across all methods, adhering to the DFME setup to ensure a fair comparison. Additionally, altering the attacker model initialization in their method results in random guess performance. Although the random guess performance of their method further underscores its limitations in real-world attack scenarios, we followed the same setup to ensure their performance remains **reasonable** for comparison. However, we agree with the reviewer and will additionally evaluate performance using pretrained weights.
> > >
> > > Note: For PASCAL only, we initialized the victim model with pretrained weights, as a randomly initialized victim model performs poorly, undermining the purpose of model stealing.

---

> > > > ### Comment · Reviewer_9Eda · 2024-11-22
> > > >
> > > > Thank you for the clarification regarding the initialization of weights. For a fair comparison with previous approaches, this makes absolutely sense.
> > > >
> > > > After checking all reviews and the authors' responses, I decided to keep my initial score. The paper is well-written, the approach offers sufficient novelty, and the results are convincing. However, I also understand some weaknesses raised by other reviewers, which is why I decided not to increase my score.
> > > >
> > > > I look forward to discussing the advantages and drawbacks of the paper with the other reviewers in the final discussion phase.

---

### Official Review · Reviewer_i8r1 · 2024-10-31

**Soundness:** 3
**Presentation:** 3
**Contribution:** 1
**Rating:** 5
**Confidence:** 3

**Summary:**

The authors propose a realistic threat model where the attacker has no access to prompts or class information from the victim model. Specifically, they introduce a novel model-stealing approach, Stealix, which leverages the CLIP model to automatically generate prompts optimized through contrastive learning. This method removes the need for human-crafted prompts.

**Strengths:**

The paper is well-written and easy to follow.
It is interesting that the design of the next triplet considers both effectiveness and diversity.
The extensive experiments and accompanying discussion provide valuable insights.

**Weaknesses:**

1. The paper employs two large foundation models to steal knowledge from a relatively small victim model, specifically ResNet-34. This approach, however, may not seem practical. If the attacker’s objective is to steal general knowledge from a smaller model, the large foundation models already possess advanced capabilities that might surpass what a smaller model like ResNet-34 could offer. Conversely, if the goal is to steal domain-specific knowledge, such as that learned in medical imaging, these large foundation models may also lack such specialized knowledge, potentially leading to lower quality generated images and reduced accuracy due to the limitations of CLIP and the diffusion model (DM). Could the authors provide a more practical scenario where their method would be beneficial? Additionally, it would be helpful to know if it’s feasible to extract knowledge from a larger model, as these models are more expensive to train and may use private datasets, making the task more realistic.

2. Although the method claims not to rely on class names or training data distribution, it does require at least one image per class. This requirement may actually provide more information than a class name alone. For instance, selecting an image per class implies prior knowledge of each category, and images include more detailed information than a simple class label. Given this, it seems that the primary novelty over previous work lies in using CLIP to automatically generate prompts rather than manually crafting them.

**Questions:**

see weakness

---

> ### Author Response · Authors · 2024-11-20
> **Response to Reviewer i8r1**
>
> We thank the reviewer for their insightful feedbacks. It is encouraging that the reviewer finds our method novel, particularly in enhancing image generation effectiveness and diversity, as well as in removing the need for human-crafted prompts. We aim to answer the concerns below.
>
> > W1.1. If the attacker’s objective is to steal general knowledge from a smaller model, the large foundation models already possess advanced capabilities that might surpass what a smaller model like ResNet-34 could offer.
>
> **While we agree that foundation models have zero-shot classification capabilities, they cannot surpass task-specific models in many general applications.** Specifically, EuroSAT dataset in our evaluation is an example where zero-shot CLIP underperforms compared to a task-specific model, as noted in the CLIP paper [1]. Other limitations are also well-documented, e.g., in OpenAI’s [blog](https://openai.com/index/clip/):
>
> 1. **Sensitivity to wording**: Achieving accurate performance with zero-shot CLIP requires careful prompt engineering.
> 2. **Limited generalization**: CLIP struggles with abstract tasks and images not seen during training, such as counting objects or estimating distances, like how close a car is in a photo.
> 3. **Challenges in fine-grained classification**: It faces difficulty distinguishing fine-grained differences, such as between car models, aircraft variants, or flower species.
>
>
>
>
> > W1.2. The lack of domain-specific knowledge in CLIP and diffusion model (DM) leads to lower performance.
>
>
> While we agree that domain-specific knowledge might be lacking in pretrained CLIP and DM, their ability to generate novel concepts might be mainly constraint by the user's ability to describe the desired target through text, as stated in Textual Inversion [2]. This statement is consistent with our method and results: with an appropriate prompt optimization objective (for us, prompt consistency), DM can significantly increase the risk of model stealing compared to previous methods.
>
>
> > W1.3. If it’s feasible to extract knowledge from a larger model?
>
> Yes, it is feasible. Our method does not rely on the victim model's size, as shown in Appendix D and E across various attacker and victim architectures and sizes. If the reviewer is referring to larger datasets rather than to larger model size, it is still feasible as the attacker increases the query budget and the surrogate model size if necessary. We are glad to address further concerns if the reviewer suggests something else.
>
>
> > W2. ...(one image per class) may actually provide more information than a class name alone ...it seems that the primary novelty over previous work lies in using CLIP to automatically generate prompts rather than manually crafting them.
>
> While one image per class might provide more information, **relying on seed images and CLIP does not guarantee effective prompts for model stealing.** When prompts are optimized solely on the seed image using CLIP, accuracy drops significantly (Tab. 5, "Stealix w/o reproduction"), e.g., from 40.0% to 26.7% in PASCAL, highlighting the importance of our work. Moreover, even with the seed image and class name, prompts generated by InstructBLIP (built on CLIP) fail to synthesize accurate EuroSAT images (Fig. 4), leading to lower attacker model accuracy (Tab. 2). The clarification of the one-image-per-class setup is in the general response [G1](https://openreview.net/forum?id=kvN8MJTOCM&noteId=dJYaD8RKq6) and our novelty in [G2](https://openreview.net/forum?id=kvN8MJTOCM&noteId=dJYaD8RKq6).
>
>
> [1] Radford, Alec, Jong Wook Kim, Chris Hallacy, Aditya Ramesh, Gabriel Goh, Sandhini Agarwal, Girish Sastry et al. "Learning transferable visual models from natural language supervision." In International Conference on Machine Learning (ICML), 2021.
>
> [2] Rinon Gal, Yuval Alaluf, Yuval Atzmon, Or Patashnik, Amit H. Bermano, Gal Chechik, and Daniel Cohen-Or. An image is worth one word: Personalizing text-to-image generation using textual inversion. In International Conference on Learning Representations (ICLR), 2023.

---

> > ### Comment · Reviewer_i8r1 · 2024-11-22
> > **Thanks for your reply**
> >
> > Thank you for the detailed response. However, my concerns still remain. While the authors cite a paper supporting the statement: "While we agree that domain-specific knowledge might be lacking in pretrained CLIP and DM, their ability to generate novel concepts might be mainly constrained by the user's ability to describe the desired target through text, as stated in Textual Inversion," I find this argument insufficient to fully address my concerns.
> >
> > I agree that detailed prompts can somewhat improve the quality of image generation for new tasks. However, I am still skeptical that a diffusion model (DM) that has not been trained on a specific dataset (e.g., medical images, as mentioned in your general response) could generate accurate and high-quality medical images purely based on prompting. If the model lacks prior exposure to specific medical concepts or professional terms, how can it meaningfully interpret and generate such images? Similarly, if CLIP has never been trained on or exposed to medical images, it is highly unlikely to produce accurate textual descriptions for such images.
> >
> > Specifically, while earlier approaches utilized text prompts, this method employs image prompts (contain more detailed information), converts them into text prompts using CLIP, and applies some optimization. This reliance on prior methodologies [1] raises concerns about the level of novelty in the proposed approach.
> >
> > For these reasons, I find myself unable to fully support the work and will maintain my current score.
> >
> > [1]Yuxin Wen, Neel Jain, John Kirchenbauer, Micah Goldblum, Jonas Geiping, and Tom Goldstein. Hard prompts made easy: Gradient-based discrete optimization for prompt tuning and discovery. (NeurIPS), 2024.

---

> > > ### Author Response · Authors · 2024-11-27
> > > **Response to Reviewer i8r1; Limited Domain Knowledge**
> > >
> > > We sincerely thank the reviewer for their time and the prompt reply. We would like to further clarify the comments below.
> > >
> > > > #### Model stealing approaches struggle when DMs lack domain-specific knowledge
> > >
> > >
> > > We want to highlight that the issue raised by the reviewer applies universally to all model stealing methods, not specifically to ours. Instead, our work demonstrates a key breakthrough: generative priors like DMs can be leveraged more effectively when they describe the data well but are not properly prompted. In other words, **our approach shares the same lower-bound as existing methods but significantly improves the upper-bound**, achieving an approximate 7–22% improvement compared with second-best method (Table 1).
> > >
> > > With that being said, we conducted an experiment analyzing performance when DMs only have limited domain-specific knowledge as concerned by the reviewer. We consider two medical datasets: PatchCamelyon (PCAM) [1] and RetinaMNIST [2]. In PCAM, class names are 'benign tissue' and 'tumor tissue'. RetinaMNIST involves a 5-level grading system for diabetic retinopathy severity, with class names as 'diabetic retinopathy $i$,' where $i$ ranges from 0 to 4 for severity. We conduct experiments using three random seeds and report the mean attacker accuracy below. The results show that methods with DM priors still outperform Knockoff and DFME, affirming the value of priors. However, the improvements decrease as the data deviates from DM's distribution, resulting in only modest gains of Stealix in such cases.
> > >
> > > |   | Victim | KD    | Knockoff | DFME | ASPKD | Real Guidance | DA-Fusion | Stealix (ours) |
> > > |------------------------|--------|-------|----------|------|-------|---------------|-----------|----------------|
> > > | PCAM                   | 91.2%  | 76.3%| 50.0%      | 50.0% | 60.1% | 61.8%         | 61.5%     | **62.2%**      |
> > > | RetinaMNIST                  | 61.7%  | 59.4%| 56.1%      | 46.1% | 55.3% | 56.1%         | 56.7%     | **58.0%**      |
> > >
> > >
> > > In summary, our approach provides
> > > 1. Significant improvement when DMs can describe the data
> > > 2. Comparable or slightly better performance when DMs have limited domain knowledge
> > >
> > > The new experiment and discussion, together with the qualitative comparison, is provided in Appendix J in the updated version.
> > >
> > > >#### Reusing Wen et al. in Stealix
> > >
> > >
> > > Wen et al. are just used as a building block for prompt optimization and are orthogonal to our contributions. Our approach is flexible and can benefit from replacing it with more advanced methods. Furthermore, the experiment in Table 5 already demonstrates that solely using Wen et al. without our proposed prompt evolution components is insufficient to effectively address the task.
> > >
> > > [1] Veeling, Bastiaan S., Jasper Linmans, Jim Winkens, Taco Cohen, and Max Welling. "Rotation equivariant CNNs for digital pathology." In Medical Image Computing and Computer Assisted Intervention (MICCAI) 2018.
> > >
> > > [2] Jiancheng Yang, Rui Shi, Donglai Wei, Zequan Liu, Lin Zhao, Bilian Ke, Hanspeter Pfister, and Bingbing Ni. "Medmnist v2-a large-scale lightweight benchmark for 2d and 3d biomedical image classification." Scientific Data, 2023.

---

> > > ### Author Response · Authors · 2024-12-02
> > > **Reminder**
> > >
> > > Dear reviewer,
> > >
> > > We have added new experiments in our latest response and in Appendix J, demonstrating that the limitation in the medical domain is common across all methods while showing our approach outperforms others within this constraint. Could you please confirm if this addresses your concern? If so, we would kindly request a reconsideration of the score.

---

### Official Review · Reviewer_LjBk · 2024-11-01

**Soundness:** 3
**Presentation:** 3
**Contribution:** 3
**Rating:** 5
**Confidence:** 3

**Summary:**

The paper introduces Stealix, a novel approach for model stealing that operates without requiring predefined prompts. The focus of the research is to expose the risk posed by pre-trained generative models in stealing proprietary machine learning models. Stealix employs two open-source pre-trained models and a genetic algorithm to iteratively refine prompts and generate synthetic images that align with the victim model’s data distribution. This enables attackers to replicate the victim model’s functionality using significantly fewer queries, even without expertise in prompt design. The approach highlights vulnerabilities in using pre-trained generative models for tasks like image synthesis, especially when coupled with black-box models available via public APIs.

**Strengths:**

1. Introduction of a novel threat model: The paper proposes a novel realistic threat model that deepens our understanding of the risks associated with pre-trained generative models.

2. Well-written and clear exposition: The writing is clear, making the technical details accessible. The paper explains its methodology and results effectively, allowing the reader to follow the progression of the experiments and findings.

3. Strong performance: The experimental results show that Stealix significantly outperforms previous methods in terms of attack model accuracy, even with a low query budget.

**Weaknesses:**

1. Prompt Refinement as the major factor in performance improvement:  In comparing the results from Table 5 with those from other methods in Table 1, it appears that the main contributor to Stealix’s performance improvement is the Prompt Refinement process. However, this refinement approach seems quite similar to that proposed by Wen et al. (2024). The paper does not clearly highlight the unique advantages of Stealix's refinement method compared to the existing technique. Furthermore, Stealix utilizes the same models (OpenCLIP-ViT/H and Stable Diffusion v2) as Wen et al. (2024) for their experiments, raising the question of what specifically distinguishes Stealix.

2. Lack of details on t in generating S: The paper could benefit from a more detailed explanation of how t is set or updated during the generation of S. For instance, in Algorithm 1, t is updated throughout the process, but it is not clear how this is done in the experiments. Providing more detail on this parameter would help clarify the methodology.

3. Missing experiments on Prompt length (L): The paper does not provide experimental results on the impact of different prompt lengths (L). A more detailed analysis of how the prompt length affects performance could improve the evaluation of the method.

4. Lack of time comparisons: The Stealix method appears to be time-consuming. Providing a report on the computational time required for Stealix compared to other methods would aid in understanding its practical applicability. In Appendix A, line 662, the authors mention "to save optimization time," but a more detailed explanation of the actual time saved or required would be beneficial. This is an essential metric for evaluating whether the proposed threat model is feasible in real-world scenarios and whether it should be prioritized in risk management.


[Reference] Yuxin Wen, Neel Jain, John Kirchenbauer, Micah Goldblum, Jonas Geiping, and Tom Goldstein. Hard prompts made easy: Gradient-based discrete optimization for prompt tuning and discovery. In Advances in Neural Information Processing Systems (NeurIPS), 2024.

**Questions:**

1. What distinguishes Stealix's Prompt Refinement from Wen et al. (2024)'s approach?: The performance improvement seems to come primarily from Prompt Refinement, but the method appears very similar to Wen et al.'s. Could you clarify what specific aspect of Stealix leads to its superior performance?

2. Could you explain in more detail what is meant by "one-shot setup" in the Method section, lines 214-215? Does the attacker need to know at least one class from the victim model, or can it work without any class information? Additionally, what are the implications of this one-shot setup on the method's practicality, and how does it compare to other approaches in terms of required prior knowledge?

3. Source code not available: The absence of source code makes it difficult to reproduce the results and assess the practicality of the method. Sharing the code would be beneficial for further verification and application.

Additionally, I would appreciate responses to the above weaknesses points.

[Reference] Yuxin Wen, Neel Jain, John Kirchenbauer, Micah Goldblum, Jonas Geiping, and Tom Goldstein. Hard prompts made easy: Gradient-based discrete optimization for prompt tuning and discovery. In Advances in Neural Information Processing Systems (NeurIPS), 2024.

---

> ### Author Response · Authors · 2024-11-20
> **Response to Reviewer LjBk**
>
> We thank the reviewer for the insightful and positive feedback. We are glad the reviewer found our method novel, our proposed threat model realistic, and the paper well-written. We address the questions below.
>
>
> > Q1. What distinguishes Stealix's Prompt Refinement from Wen et al. (2024)'s approach? The performance improvement seems to come primarily from Prompt Refinement.
>
>
> The main difference lies in the use of triplet images to capture features learnt by the victim model. Specifically, Wen et al. (2024) optimize prompts with the seed image $x_c^s$, while ours integrates the victim model’s predictions with a triplet $(x_c^s, x_c^+, x_c^-)$ in the prompt refinement. This highlight positive features and discard negatives learnt by the victim model. We clarify that 'Stealix w/o reproduction' in Table 5 refers to using Wen et al.'s method with only $x_c^s$ to train the attacker model. Their method (Tab. 5, Stealix w/o reproduction) results in a low performance (e.g. 26.7% on PASCAL) compared with ours (40.0%), demonstrating the effectiveness of our prompt refinement and reproduction.
>
>
>
> > Q2. Could you explain in more detail what is meant by "one-shot setup" in the Method section, lines 214-215? Does the attacker need to know at least one class from the victim model, or can it work without any class information? Additionally, what are the implications of this one-shot setup on the method's practicality, and how does it compare to other approaches in terms of required prior knowledge?
>
> We borrow this terminology from existing literature, such as DA-Fusion [1], which refers to a setting where there is **one image per class** ($|X_c^s|=1$) available. In practice, end-users, like competitors, often have limited images at hand, which justifies the setting and motivates the attack. For additional clarification on why this setting is realistic, we refer the reviewer to general response [G1](https://openreview.net/forum?id=kvN8MJTOCM&noteId=dJYaD8RKq6). In Table 1, we compare our approach closely with various baselines, including DA-Fusion and Real Guidance, both of which also rely on the one-image-per-class setting.
>
> [1] Brandon Trabucco, Kyle Doherty, Max Gurinas, and Ruslan Salakhutdinov. Effective data augmentation with diffusion models. In International Conference on Learning Representations (ICLR), 2024.
>
>
> > Q3. Source code not available.
>
> We are committed to making the code publicly available upon acceptance, as noted in the Reproducibility Statement.
>
> > W1. Lack of details on t in generating S in Algorithm 1. The paper could benefit from a more detailed explanation of how t is set or updated during the generation of S.
>
> $t$ is the population index, as defined in line 3 and updated in line 23 of Algorithm 1.
>
>
> > W2. Missing experiments on Prompt length (L).
>
>
> Thanks for the suggestion. We additionally evaluate the impact of different prompt lengths (4, 16, and 32) on EuroSAT, using a query budget of 500 per class across three random seeds. The table below presents the mean accuracy of the attacker model. Stealix consistently outperforms other methods (best baseline is 59.0% from DA-Fusion) across all prompt lengths. The observed differences in performance further justify the choice with prompt length 16, balancing efficiency with accuracy.
>
> | Prompt length | 4             | 16            | 32            |
> |---------------|---------------|---------------|---------------|
> | Stealix       | 62.5%  | 65.9% | 64.3%  |
>
>
> > W3. Lack of time comparisons... In Appendix A, line 662, the authors mention "to save optimization time," but a more detailed explanation of the actual time saved or required would be beneficial.
>
> We appreciate the suggestion and provide a time comparison with EuroSAT as an example. All experiments were run on a single machine equipped with an NVIDIA V100 32GB GPU and an AMD EPYC 7543 32-Core Processor (Line 320-321).
>
> For the prompt refinement, each prompt requires ~18 seconds for 500 optimization steps, compared to ~3 minutes for 5000 steps.
>
> Additionally, we present the time taken for the entire process, using a 500-query budget per class (DFME uses 2M queries per class). The table below shows that Stealix achieves state-of-the-art accuracy with a reasonable computational time.
>
> |                | Knockoff | DFME | ASPKD | Real Guidance | DA-Fusion | Stealix (ours) |
> |----------------|----------|------|-------|---------------|-----------|----------------|
> | **Time (hours)** | 0.5     | 4.5  | 28.6  | 3.3           | 5.4       | 6.3            |
> | **Accuracy (%)** | 40.1    | 11.1  | 39.2  | 51.2          | 59.0       | 65.9            |

---

> ### Author Response · Authors · 2024-11-27
> **Response to Reviewer LjBk; Revised Paper**
>
> We have updated the paper with the suggestions made by the reviewer: Impact of prompt length in Appendix A marked as blue; "Comparison of Computation Time" in Appendix I.
>
> If the concern has been properly addressed, we would kindly request the reviewer to reconsider the score.

---

### Official Review · Reviewer_85xe · 2024-11-01

**Soundness:** 2
**Presentation:** 2
**Contribution:** 2
**Rating:** 6
**Confidence:** 3

**Summary:**

The paper proposes Stealix, a new method to steal black-box classification models with the help of open-source models. Stealix iteratively (1) optimizes prompts so that they can produce images that are close to the images from the target class using the open-source text-to-image models, (2) evaluates the prompts with prompt consistency score defined as the fraction of the generated images that are labeled as the target class by the victim model, and (3) use the optimized prompts and their generated images to construct the image set for the next iteration. Finally, Stealix trains a classifier using the collected samples from the above process. Experiments on EuroSAT, PASCAL, CIFAR10, and DomainNet show that the proposed algorithm can train a surrogate that achieves better classification accuracy than prior approaches.

**Strengths:**

* The paper evaluates the proposed algorithm on 4 datasets, and the results look promising.

**Weaknesses:**

* My main concern is that the description of the method sections is quite confusing. It makes it hard for me to fully understand every detail of the algorithms and judge their validity.

* The paper studies the setting where the class names of the victim are unknown to the attacker, while one image per class is available to the attacker. The paper claims that this setting is realistic, but that is questionable to me.

Please see the "Questions" section for more elaboration of these points.

**Questions:**

## The description of the method is confusing

*  Fitness value in Section 4.3 is defined for prompts, not images. I am confused about the description in Section 4.4 "The triplet with the highest fitness value (PC) in S t is selected as the elite, carried forward to the next generation S t+1to guide the production of improved triplets." The triplet by definition contains 3 images, with no prompts inside. How do you define the fitness value for the triplet?

* In Section 4.4 it says "Once the parent set is formed, two parents from S p are chosen sequentially, each split at a random point. The segments are recombined to form a new triplet, ensuring both parents contribute to the new triplet." The two parents are just two triplets, each with 3 images. What do you mean by "split at a random point" and "segments" for image triplets?

* Will the negative samples in line 25 of Algorithm 1 be used to train the surrogate model? If so, do you use their raw label from the victim model as the training label, or simply minimize their prediction prob of class c? This is not described in the paper.

* Are the prompts randomly initialized at every iteration t?

*  Line 4 of the algorithm says that the last two elements of the image triplets are randomly sampled from the sets X_c^+ and X_c^-. But these two sets are initialized as empty sets in Line 3. How do you draw samples from empty sets?

* Line 16 of Algorithm 1 claims that Line 17 and Line 18 consume the query budget. But the budget should have already been consumed in Line 15.

* What does "CIFAR10 is a standard classification dataset, where class names can be treated as ground truth for synthesizing relevant images." mean in Section 5.1? Do you mean that you use the class names in CIFAR10 experiments? If so, it contradicts the earlier claims in the paper.

* Line 460 says "retain the positively labeled images to form the transfer dataset for training the attacker model.". Could the authors elaborate on it more? I do not understand this sentence.

## The studied setting is questionable

The paper studies the setting where the class names of the victim are unknown to the attacker, while one image per class is available to the attacker, and claims that this setting is more realistic than prior work. However, I have some questions:

* **Regarding class names.** The paper claims that the assumption that the attacker does not know class names made in this paper is a significant advance compared to prior work. However, if the service provider of the classification model does not release the class names publicly, then this classification model has no use to the end users, as the users do not know what the returned classification results mean. All the classification services out there I am aware of have released the class information publicly. Could the authors provide an example of a real-world classification service where class name information is not known to end users?

* **Regarding one image per class.** The paper assumes that one image per class is available to the attacker. I would regard this as a huge constraint. I can think of real-world cases where getting those images is hard, such as medical image diagnosis systems. Moreover, assuming "one image per class is available, while class names are unavailable" is unnatural to me. If the attacker has access to one image per class, they can directly look at those images and might be able to have a good guess of the class name. Again, it would be more convincing if the authors could provide real-world examples where such assumptions hold.

* **Regarding the experiments.** The paper should list clearly which baseline methods require real images (and how many) and which do not. Without noting that, it is hard to say if the claimed benefit of Stealix over baseline methods comes from the availability of real images (one image per class) or the designed methodology.

Moreover, given the availability of one image per class, one simple approach is to augment the images with approaches such as DA-Fusion, and then train the surrogate model directly. If I understand it correctly, such a simple baseline is not considered in the paper.


## Other questions

* Line 16 of Algorithm 1: consumed -> consume

* Line 446 claims that "Since the attacker lacks access to victim data", but the paper assumes that one image per class is available to the attacker.

* The paper claims in multiple places that their generated data distribution aligns with that of the victim data, such as "By designing prompts to synthesize images that align with the victim data distribution, ..." in Section 4.1,  "we underscore the crucial role of aligning the distribution of generated data with that of the target model" in Section 7, and many more (just search "align" and you will find all the places). Note that "align the distribution" has a mathematical implication and it can mislead the readers to think that you are minimizing d(p_victim, p_synthetic), where p_real and p_synthetic are the distributions of victim data and synthetic data, for some distance metric d. Obviously, the proposed approach is not doing that (or at least, this claim is not quantitatively evaluated and justified). I would suggest the authors to rephrase these claims.

------
Given all these concerns, I have to give a rejection. If I missed anything, please correct me in the rebuttal and I will be happy to revisit the score.

**Details Of Ethics Concerns:**

The paper did a good job of discussing the ethical concerns in the ethics statement. However, given that attackers can use the proposed approach to steal black-box models, an ethics review might be needed.

---

> ### Author Response · Authors · 2024-11-20
> **Response to Reviewer 85xe; Part 1**
>
> We thank the reviewer for the thoughtful feedback and considering our results promising. We now address the concerns below.
>
> > Q1. Regarding class names... Could the authors provide an example of a real-world classification service where class name information is not known to end users?
>
> We clarify that we do not mean that class names are unavailable, but might be insufficient to generate images similar to the victim data, such as EuroSAT shown in Fig. 4 and 5. We limit class name use just because they may by chance serve as good prompts for dataset like CIFAR10, while we allow baselines like Real Guidance to use both class names and seed images, demonstrating our approach outperforms others across threat models.
>
> > Q2. Regarding one image per class ... I would regard this as a huge constraint...such as medical image diagnosis systems.
>
> Model stealing attack could be launched by end-users like **competitors** within the same field, as discussed in general response [G1](https://openreview.net/forum?id=kvN8MJTOCM&noteId=dJYaD8RKq6). For instance, in the medical domain, competitors may have limited medical images, but restrictions like high costs and privacy hinder their scalability, driving them to steal models with the limited images.
>
> > Q3. ... If the attacker has access to one image per class, they can directly look at those images and might be able to have a good guess of the class name.
>
> As mentioned in Q1, the class names are available but likely fail to describe the victim data accurately. Moreover, prompt engineering is labor-intensive and non-trivial, which often limits scalability (L15-16). In contrast, our approach provide an automatic and scalable solution. To further investigate the limitations of prompt engineering, we adopt cutting-edge vision-language model, InstructBLIP, in analysis "Limitations of human-crafted prompts" (Line 374). Given the class name and seed image, the prompt designed by InstructBLIP fails to synthesize EuroSAT images in Fig. 4. It indicates that seed images associated with their class names are not sufficient to synthesize victim data, which necessitates our approach.
>
> > Q4. The paper should list clearly which baseline methods require real images (and how many) and which do not.
>
> Table 1 has listed the requirements of different approaches, including the number of real images and the use of class names. We will explicitly highlight this for better clarity.
>
> > Q5. ... augment the images with approaches such as DA-Fusion, and then train the surrogate model directly.
>
> While data augmentation without querying the victim model is not considered as model stealing, we include an additional experiment comparing attacker model accuracy in model stealing and data augmentation setup suggested by the reviewer (last line) with 500 query budget per class. The table below shows that performance degrades significantly with DA-Fusion when relying solely on class labels for training, highlighting that model stealing is essential, even with one image per class.
>
> | **Method**                             | **Query victim** | **EuroSAT**          | **PASCAL**           | **CIFAR10**          | **DomainNet**        |
> |----------------------------------------|------------------|----------------------|----------------------|----------------------|----------------------|
> | **Victim**                             | -                | 98.2%      | 82.7%        | 93.8%        | 83.9%        |
> | **Stealix**                     | ✓               | **65.9%**   | **40.0%**    | **49.6%**     | **48.4%** |
> | **DA-Fusion**       | ✓                | 59.0%        | 16.4%        | 26.7%       | 28.4%       |
> | **DA-Fusion**      |   ✗             | 29.9%       | 10.7%         | 18.9%       | 17.9%         |
>
> > Q6. ... How do you define the fitness value for the triplet?
>
> The fitness value of a triplet is the value of its associated prompt, which is derived through prompt refinement with triplet images (Equation 3).
>
> > Q7. ... What do you mean by "split at a random point" and "segments" for image triplets?
>
> It means randomly exchanging images between two triplet sets to create a new set. For example, given two triplets $(x_1, x_2, x_3)$ and $(x_4, x_5, x_6)$, a new triplet could be like $(x_1, x_5, x_6)$. We will update the phrasing for clarity.
>
> > Q8. Will the negative samples in line 25 of Algorithm 1 be used to train the surrogate model? If so, do you use their raw label from the victim model as the training label, or simply minimize their prediction prob of class c? This is not described in the paper.
>
> Yes, they are used to train the surrogate model, as stated in Line 213-215, where $B$ contains both positive and negative images. We use the raw label from the victim model, as described in the Equation (2). We will highlight this in the paper.

---

> ### Author Response · Authors · 2024-11-20
> **Response to Reviewer 85xe; Part 2**
>
> > Q9. Are the prompts randomly initialized at every iteration t?
>
> Yes, they are. As shown in Algorithm 1, Line 12, the **PromptRefinement** function does not accept previous prompts. This initialization is also stated in Line 3 of Algorithm 2.
>
> > Q10. Line 4 of the algorithm says that the last two elements of the image triplets are randomly sampled from the sets X_c^+ and X_c^-. But these two sets are initialized as empty sets in Line 3. How do you draw samples from empty sets?
>
>
> Sampling from empty sets means no samples are selected, thus the population starts with only seed images.
>
> > Q11. Line 16 of Algorithm 1 claims that Line 17 and Line 18 consume the query budget. But the budget should have already been consumed in Line 15.
>
>
> We clarify that Lines 17 and 18 **update the sets**, and Line 19 **updates the variable** that tracks the consumed budget, instead of **consuming the query budget**, as they are consumed in Line 15.
>
>
>
> > Q12. What does "CIFAR10 is a standard classification dataset, where class names can be treated as ground truth for synthesizing relevant images." mean in Section 5.1? Do you mean that you use the class names in CIFAR10 experiments? If so, it contradicts the earlier claims in the paper.
>
> No, our method does not use CIFAR-10 class names or any other class names. We mean that class names of CIFAR10 can sufficiently act as a guide for relevant image synthesis, thus methods like Real Guidance using class names of CIFAR10 is a strong baseline.
>
>
>
> > Q13. Line 460 says "retain the positively labeled images to form the transfer dataset for training the attacker model.". Could the authors elaborate on it more? I do not understand this sentence.
>
> We only use positive images ($X_c^+$) to train the attacker model in this analysis, differing from the main setting (Algorithm 1). This experiment examines how maximizing the PC improves attacker model performance. For instance, the victim model classifies images prompted with "black dog" and "white dog" as "dog," but "white dog" has a higher PC. We demonstrate that higher PC enhances model performance (see Fig. 6). Using only positive images ensures that 'dog' images are generated strictly from 'dog' prompts (e.g., 'black dog' or 'white dog') rather than unrelated prompts for other classes like 'white cat.'
>
> > Q14. Line 446 claims that "Since the attacker lacks access to victim data", but the paper assumes that one image per class is available to the attacker.
>
> One image is not the same as a full dataset or distribution in terms of realism and feasibility for the attacker. To improve clarity, we will update this claim to: 'Since the attacker lacks access to the distribution of the victim data.'
>
>
> > Q15. The paper claims in multiple places that their generated data distribution aligns with that of the victim data ... Note that "align the distribution" has a mathematical implication and it can mislead the readers to think that you are minimizing it ... I would suggest the authors to rephrase these claims.
>
> Thanks for the suggestion, we will replace the use of this term.

---

> > ### Comment · Reviewer_85xe · 2024-11-25
> > **Thank you!**
> >
> > Thanks for the detailed rebuttal! The authors did a good job of addressing my questions.
> >
> > Re: Q1, Q2, Q3. Thanks for the explanations. I am convinced by your justification of the problem setting now, especially the setting where an organization wants to steal the model from competitors.
> >
> > Re: Q4. Sorry that I missed these descriptions when I reviewed the paper.
> >
> > Re: Q5. Thanks for adding the experiments!
> >
> > Re: Q6, Q7, Q8, Q9, Q11, Q12, Q13, Q14, Q15. Thanks for the explanations.
> >
> > Re: Q10. If we "start with only seed images", how do we execute PromptRefinement in line 12? I still do not fully understand how the first iteration works.
> >
> > Given that most of my concerns are addressed, I will increase the score to 5 now. The reason I do not increase the score to 6 is that,  for Q1, Q2, Q3, Q5, Q6, Q7, Q10, Q11, Q12, Q13, Q14, Q15, the writing of the paper needs to be updated to justify the problem setting better or improve the clarity of the method/experiment descriptions. These require quite a lot of changes and deserve a careful check by the reviewers. However, I do not see a revision uploaded by the authors.
> >
> > In addition, I need to mention that, I do not work in the field of model stealing so I am not able to give a fair and good judgment on the novelty of the work. The score and evaluation I give do not take novelty into account. I would defer this aspect to other reviewers with better expertise.

---

> > > ### Author Response · Authors · 2024-11-27
> > > **Response to Reviewer 85xe; Revised Paper**
> > >
> > > Thanks for increasing the score! We didn't update the paper earlier, as reviewers use line numbers to reference questions. Since we've addressed most concerns, we have just uploaded the revised version. The main updates are marked as blue in the paper. We provide here an overview of the updates:
> > >
> > > > Q10: If we "start with only seed images", how do we execute PromptRefinement in line 12?
> > >
> > > We optimize the prompt by minimizing the  loss on the triplet of images $\mathbb{X}^{s+-}_c = \\{x_c^s, x_c^+, x_c^-\\}$, such that (Line 730):
> > >
> > > $$ \min_p\sum_{\mathbf{x} \in \mathbb{X}^{s+-}_c} \mathcal{L}(I(x), T(p), V(x))$$
> > >
> > > If we start with only the seed image, then $\mathbb{X}^{s+-}_c$ is replaced with  $\mathbb{X}^{s}_c = \\{x_c^s\\}$ in the loss function.
> > > We have updated Equation 3 in Line 227 for better clarity.
> > >
> > > > Q1.Q2.Q3. Clarify why we limit class name
> > >
> > > Update Line 44-47 in Introduction: Previous approaches use human-crafted prompts or class names to synthesize images with a text-to-image model, but they overlooks scenarios where class names lack context or fail to capture victim data features. Attackers may also lack sufficient knowledge of the victim's data distribution or may struggle to describe it accurately.
> > >
> > > Update Line 153-155 in Threat Model: We also limit the use of class names, as they may by chance serve as good prompts; using them would diverge from the assumption that the attacker lacks prompt design expertise.
> > >
> > > > Q5: DA-Fusion as data augmentation
> > >
> > > Provided in Appendix H (Line 918),
> > >
> > > > Q6: Fitness for triple images.
> > >
> > > Update Line 258: Since the prompt is optimized with a triplet of images, the fitness value can also be assigned to the corresponding triplet in $\mathcal{S}^t$
> > >
> > > > Q7: Clarify "split at a random point" in prompt reproduction
> > >
> > > Update Line 269: Once the parent set is formed, two parent triplets are selected, and their images are randomly exchanged to create a new triplet, ensuring contributions from both parents.
> > >
> > > > Q11: Update the consumed budget
> > >
> > > Update Code line 17 in Algorithm 1: Code $b \gets b+M$ is put right after the budget is consumed for better clarity.
> > >
> > > > Q12: Clarify CIFAR10 class names are not used in our method.
> > >
> > > Update Line 314: In CIFAR10, class names can guide image synthesis, leading to strong baselines when used by other methods, compared to ours, which does not.
> > >
> > > > Q13: Only use positive images in one analysis experiment
> > >
> > > Update Line 459: "retain the positively labeled images to form the transfer dataset for training the attacker model" -> "use only the positive
> > > images to train the attacker model."
> > >
> > > > Q14: Clarify the attacker lacks access to victim data distribution.
> > >
> > > Update Line 431: "Since the attacker lacks access to victim data" -> "Since the attacker lacks access to the distribution of the victim data"
> > >
> > > > Q15: Avoid using "align with the victim data distribution"
> > >
> > > We've replaced this statement throughout the paper with phrases like 'synthesize images similar to the victim data.' Reviewers can verify this by searching for 'align', which we don't use anymore.
> > >
> > > We wish these updates address the last concern of the reviewer and appreciate that if the reviewer could increase the score to 6.

---

> > > ### Author Response · Authors · 2024-12-02
> > > **Reminder**
> > >
> > > Dear reviewer,
> > >
> > > we have updated the paper to improve clarity. As the rebuttal deadline approaches, could you please confirm if this addresses your concern? If it does, we kindly request your reconsideration of the score.

---

> > > > ### Comment · Reviewer_85xe · 2024-12-02
> > > >
> > > > Sorry for the late reply, and thanks for the revision.
> > > >
> > > > I have gone through the revision, and it addresses all my previous questions.
> > > >
> > > > Re: Q11. I initially misunderstood that the comment "consumed query budget" refers to lines 17 and 18 of Algorithm 1 in the original submission. That's why I thought it was an error. Looking at your explanation of the revision, I realized that it actually referred to line 19 of Algorithm 1 in the original submission. One way to help eliminate such confusion from readers would be to move the comment "Update the consumed query budget" to be between line 18 and line 19 of Algorithm 1 in the original submission. The updated algorithm in the revision is also OK to me.
> > > >
> > > > Given that my questions are addressed, I will increase the score to 6.

---

> > > > > ### Comment · Reviewer_85xe · 2024-12-02
> > > > >
> > > > > I also updated the presentation score from 1 to 2 based on the revision.

---

### Author Response · Authors · 2024-11-20
**Response to all reviewers**

# General Response

We thank all the reviewers for their time and valuable feedback. We are pleased that our results are recognized as promising (85xe, LjBk, 9Eda), our method as novel (LjBk, i8r1, 9Eda), and the threat model as realistic (LjBk, i8r1). We also appreciate the positive comments that our writing is easy to follow (i8r1, LjBk, 9Eda). Our work is the first to explore the potential of pre-trained generative models through prompt optimization, revealing a greater risk than previously recognized. We aim to provide a general clarification on the threat model, novelty of the method here.

## G1. Realistic threat model

**Our threat model targets scenarios where the class name is insufficient to generate images similar to the victim data, rather than unavailable in the real world.** We limit class name use because they may by chance serve as good prompts for datasets like CIFAR10, while we allow baselines to use for comparison. Our method significantly outperforms baselines (Tab. 1), demonstrating greater efficiency across threat models even in CIFAR10.

**Having one image per class is a realistic setup and differs from having full access to victim data or its distribution.** This reflects real-world threats posed by **competitors** in the same field, aiming to provide similar services. For instance, in the medical domain, competitors may have limited medical images, but restrictions like high costs and privacy hinder their scalability, driving them to steal models with the limited images. Such threats have become a concern for service providers. We also confirmed with an industry partner, which caters to a larger user network, that our threat model is interesting and relevant.

## G2. Novelty of the method
Our method is original and specifically designed for model stealing, rather than merely integrating existing approaches. **Each component addresses a unique challenge unexplored in previous works.**

1. **Prompt refinement**: For the first time, we integrate victim model's predictions into prompt optimization, differentiating itself from prior work and solving a novel formulation of model stealing when expertise in prompt design is unavailable.
2. **Prompt consistency**: Since measuring the quality of synthetic images in a label-only access is challenging, we propose prompt consistency as a proxy to guide prompt optimization.
3. **Prompt reproduction**: Our novel application of generic algorithms enables prompt optimization through non-differentialable signals and encourages more diverse syntehtic images.

Additionally, leveraging CLIP and the diffusion model can be viewed as a strength: our method benefits from advancements in foundation and generative models. Beyond the technical contribution, we also introduce a more realistic problem setting, as stated above in G1.

---

### Meta-Review · Area_Chair_dqgn · 2024-12-23

**Metareview:**

The submission "Stealix: Model Stealing via Prompt Evolution" proposes the use of large open-source generative models to synthesize samples to use for model stealing attacks. The submission proposes a prompt generation and refinement pipeline to generate better test images to use for model stealing, leading to an attack that improves upon previous model stealing works.

Reviewer response to this approach was relatively muted, we do think that this attack is valid, and forms some improvement over previous work, but I cannot help but think that this is a pretty heaver hammer wielded for model stealing, limiting this attacks usefulness. Both  OpenCLIP-ViT/H and Stable-Diffusion-v2 are massive pretrained models, which which it makes only limited sense to fish for CIFAR-10-like samples. Running this experiment with comparatively weaker models available to the attacker, or, instead, attacking state-of-the-art models, at least on ImageNet (or even modern image classification APIs) would have made this into a better comparison.

Reviews bring up that the reverse would also be a problem: It is not so clear from this work whether open-source models with substantially bad performance on the target task  (for example medical data was discussed) are useful enough to generate synthetic images in that task. Yet if they were, the open-source models themselves would be a better starting point. The authors point out that their approach still results in better outcomes than previous model stealing approaches, but I do not consider this a sufficient bar to clear.

In summary, I do think that this conundrum regarding the relative strength of the used large-scale open-source models is not well addressed by the current submission and wish the authors would take more time to answer this question and update their presentation based on their findings. As such, I am not recommending acceptance for now.

**Additional Comments On Reviewer Discussion:**

There were a number of issues with the clarity of presentation that the reviewers brought up during the discussion with the authors that were addressed by the authors.

---

### Decision · Program_Chairs · 2025-01-22

Reject